# The role of intra-guild indirect interactions in assembling plant-pollinator networks

Sabine Dritz [1] ✉, Rebecca A. Nelson[1] & Fernanda S. Valdovinos [1] ✉

Understanding the assembly of plant-pollinator communities has become critical to their conservation given the rise of species invasions, extirpations, and species' range shifts. Over the course of assembly, colonizer establishment produces core interaction patterns, called motifs, which shape the trajectory of assembling network structure. Dynamic assembly models can advance our understanding of this process by linking the transient dynamics of colonizer establishment to long-term network development. In this study, we investigate the role of intra-guild indirect interactions and adaptive foraging in shaping the structure of assembling plant-pollinator networks by developing: 1) an assembly model that includes population dynamics and adaptive foraging, and 2) a motif analysis tracking the intra-guild indirect interactions of colonizing species throughout their establishment. We find that while colonizers leverage indirect competition for shared mutualistic resources to establish, adaptive foraging maintains the persistence of inferior competitors. This produces core motifs in which specialist and generalist species coexist on shared mutualistic resources which leads to the emergence of nested networks. Further, the persistence of specialists develops richer and less connected networks which is consistent with empirical data. Our work contributes new understanding and methods to study the effects of species' intra-guild indirect interactions on community assembly.

Global change is driving novel assemblages of ecological communities through species invasions, extinctions, and range shifts[1,2]. Among those perturbed communities are plant-pollinator networks, which support terrestrial biodiversity and pollination services to crops[3–5]. Understanding and predicting the emergent structure and dynamics of novel plant-pollinator networks is critical to anticipating how global change will affect the ecosystem services these communities provide[6–8]. Both direct interactions between plant-pollinator pairs and indirect interactions within guilds contribute to colonizers' establishment[9–11]. Indirect interactions within a guild (pollinators or plants) occur when multiple species share the benefits of a mutualistic partner. By sharing mutualistic resources, species can have indirect competitive or facilitative effects on one another[12,13]. Over the course of assembly, colonizer establishment produces core interaction

patterns, called motifs[14,15] which function as the building blocks of ecological networks[15,16].

While the assembly of mutualistic communities is often driven by direct interactions between well-connected species[17–19], few studies have investigated the role of indirect interactions[11,20]. Indirect effects are difficult to detect empirically[13,21,22] because they require increasing the ecological and temporal scales of study in order to consider more species and interactions[23,24]. Nevertheless, indirect interactions, that propagate through short or long paths, are critical to the complexity and biodiversity of mutualistic networks[12,25–28]. Previous studies have used mathematical methods to determine the strength of indirect effects by accounting for positive and negative feedbacks[25–28]. Network motifs function as a complimentary tool to those methods by establishing a connection between indirect effects and network structure[14].

[1]Department of Environmental Science and Policy, University of California Davis, 350 East Quad, Davis, CA 945616, USA. ✉e-mail: sjdritz@ucdavis.edu; fvaldovinos@ucdavis.edu

The trade-off is that considering indirect effects through longer paths requires larger and more complex motifs which are more difficult to interpret. For simplicity, here we only consider indirect effects among species sharing a direct mutualistic partner (a path of length two). Particularly in communities where plant-pollinator mutualisms are obligate, species within a guild sharing mutualistic resources will boost or hinder each other's reproductive success[13,22,29–33].

Numerous models have been developed for the study of food web assembly[34,35] categorized as static or dynamic. Static assembly models (e.g., directed random graphs, the cascade model, the niche model) encode simple rules for the attachment of colonizers to networks[34–37]. These static models have been successful at generating structures consistent with empirically observed food webs. Several static assembly models have also been developed for mutualistic networks. These models, based on either preferential attachment to abundant generalists[19,38] or trait compatibility[39,40], have reproduced nestedness. Nestedness is a feature of plant-pollinator networks in which a core of generalists interact with generalist and specialist species while specialists mostly interact with generalist species[41]. Both topics − the emergence and the dynamic stability of nestedness − have been widely debated[38,41–43], and to date they have been studied separately.

Dynamic assembly models allow network structure to emerge and evolve with population dynamics to highlight the trajectory rather than the endpoint of assembly[34,35,44–51]. In these models, assembly is performed through a series of colonization or speciation events, while species turnover is governed by a separate population dynamics model. Dynamic assembly models developed for food webs found that complex community structure arises only when networks maintain variability in niche breadth rather than trending towards uniform generalism or specialism[51]. We are aware of only one dynamic assembly model developed for mutualisms[52]. Becker et al.'s assembly model[52] produced nested networks at intermediate stages of assembly but ultimately resulted in non-nested networks composed of only specialists in the absence of demographic noise. This is because specialist pollinators extracted resources most efficiently and excluded indirect generalist competitors within their guild. In the presence of demographic noise, Becker et al.'s assembly model[52] maintained variability in niche breadth but still resulted in non-nested networks. Here, we develop a dynamic assembly model from a consumer-resource model of plant-pollinator population dynamics that accounts for adaptive foraging by pollinators[53]. That is, the pollinators' capability to

behaviorally increase their foraging effort on the plant species in their diet with the most floral rewards available. Adaptive foraging strongly influences plant-pollinator community dynamics by partitioning pollinators' niches and providing higher quality visits to specialist plants[54–56].

Our contribution investigates the role of intra-guild indirect interactions and adaptive foraging in shaping the structure of assembling plant-pollinator networks. We do so by evaluating how colonizers' intra-guild indirect interactions influence core motif development in dynamic assembly models with and without adaptive foraging. Specifically, we ask the following questions regarding colonizers that successfully establish in the network: (1) How do intra-guild indirect interactions affect colonizer establishment? (2) Do colonizers competitively exclude intra-guild indirect specialists? (3) Do colonizers facilitate the establishment of subsequent colonizing intra-guild indirect specialists? (4) What are the core motifs characterizing assembling networks? (Fig. 1)

## Results
### Overview
We developed two dynamic assembly models, one including adaptive foraging and one excluding it, and performed a series of network assembly simulations for each. Each simulation varied in the values of two parameters corresponding with the probability that colonizing plants or pollinators will be specialist (11 probabilities for each guild ranging from 0 to 1, $11 \times 11 = 121$ simulations per model). A simulation begins with an empty network in which three plant and three pollinator species are introduced at low abundances (hereafter attempted colonizers) every 2000 timesteps for a total of 50 colonization events. Attempted colonizers are attached randomly to the network given their niche breadth type (specialists or generalists). Specialists are introduced with degree (i.e., number of interactions) one and generalists are introduced with degree drawn randomly from a uniform distribution between two and the maximum number of species in the opposite guild. This split in degrees reflects qualitative differences between specialist and generalist species in the model, which can be generalized to specialist and generalist species in empirical systems: specialist plants offer the most exclusive floral rewards and specialist pollinators perform the highest quality of pollination services. However, specialists of both guilds are vulnerable to disturbance due to more inflexible niches. Each colonizer is also assigned values for each

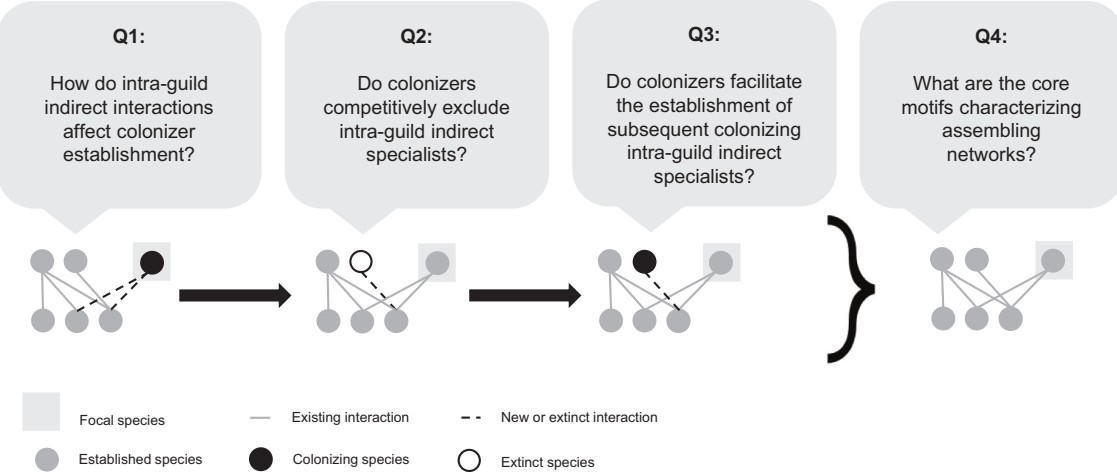

**Fig. 1 | Motif analysis to answer our guiding questions.** Each of our four guiding questions corresponds with a stage in core motif development (sequence of drawings). We developed four motif groups (see Fig. 3) to track how colonizers' intra-guild indirect interactions influence colonizers' establishment (Q1), whether colonizers competitively exclude (Q2) or facilitate the establishment (Q3) of intra-

guild indirect specialists, and core motifs produced by this process (Q4). We answer these questions for both plants (Fig. 6) and pollinators (Fig. 8). Drawings represent motifs where nodes in the same row indicate species of the same guild (pollinators or plants).

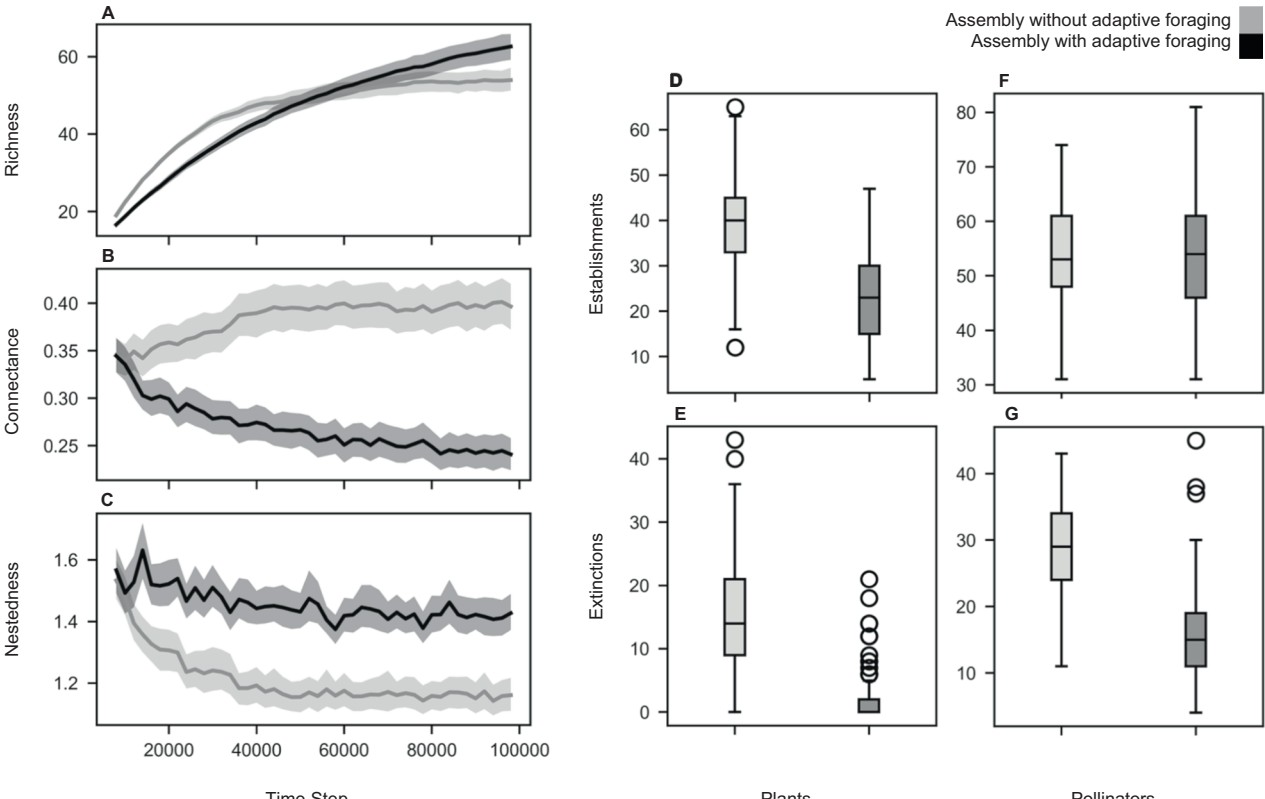

**Fig. 2 | Simulation overview.** Panels **A–C** show the trajectory of network structure over the course of assembly for the model with adaptive foraging (dark gray) and the model without adaptive foraging (light gray). The lines indicate the average value across all 121 networks and the shaded regions indicate the bootstrapped 95% confidence intervals. Panels **D** and **E** show the number of plant species' establishments and extinctions, respectively, per each of the 121 simulations for each model. The center line of the boxplots denotes the median which for plant establishments is 40 and 23 for the models without and with adaptive foraging, respectively, and for plant extinctions is 14 and 0 for the models without and with adaptive foraging, respectively. The shaded box denotes the interquartile range (IQR) which for plant establishments is [33, 45] and [15, 30] for the models without and with adaptive foraging, respectively, and for plant extinctions is [9, 21] and [0, 2] for the models without and with adaptive foraging, respectively. The whiskers denote the range of data excluding outliers which for plant establishments is [16, 63] and [5, 47] for the models without and with adaptive foraging, respectively, and for plant extinctions is [0, 36] and [0, 5] for the models without and with adaptive foraging, respectively. The points denote outliers in the data, the complete range of data including outliers for plant establishments in the model without adaptive foraging is [12, 65] and for plant extinctions is [0, 43] and [0, 21] for the models without and with adaptive foraging, respectively. Panels **F** and **G** show the number of pollinator species' establishments and extinctions, respectively, per each of the 121 simulations for each model. The median pollinator establishments is 53 and 54 for the models without and with adaptive foraging, respectively, and for pollinator extinctions is 29 and 15 for the models without and with adaptive foraging, respectively. The IQR for pollinator establishments is [48, 61] and [46, 61] for the models without and with adaptive foraging, respectively, and for pollinator extinctions is [24, 34] and [11, 19] for the models without and with adaptive foraging, respectively. The range of data excluding outliers for pollinator establishments is [31, 74] and [31, 81] for the models without and with adaptive foraging, respectively, and for pollinator extinctions is [11, 43] and [4, 30] for the models without and with adaptive foraging, respectively. The complete range of data including outliers for pollinator extinctions in the model with adaptive foraging is [4, 45].

parameter in our model (see Methods) by sampling a uniform distribution of a given mean and variance (Table S1). Only a proportion of attempted colonizers successfully establish in the network (hereafter established colonizers), and between each colonization event the model reaches a steady state.

Considering the doubling time of the population dynamic model given the parameter values used, the $10^5$ timesteps that each of our simulations take corresponds to approximately 144 generations of pollinators (see Methods). Given that pollinators commonly reproduce annually, each simulation roughly spans 144 years and colonization events occur every 3 years. This timescale reasonably corresponds to novel species assemblages forming as a result of species invasions, extinctions, and range shifts – rather than considering the assembly of a community over the course of a season or over longer evolutionary timescales.

Binary interaction networks produced by the assembly model with adaptive foraging were significantly richer (Fig. 2A), less connected (Fig. 2B), and moderately more nested (Fig. 2C) than networks produced by the assembly model without adaptive foraging. In

addition, networks produced by the assembly model with adaptive foraging experienced fewer plant colonizer establishments, plant extinctions, and pollinator extinctions per simulation (Fig. 2D, E, G). The number of pollinator colonizer establishments was not significantly different between the assembly model with and without adaptive foraging (Fig. 2F). Moreover, the networks produced by our assembly model with adaptive foraging exhibited similar levels of richness, connectance, nestedness, pollinator to plant species ratio, and degree distributions as the ones observed in empirical networks (see Fig. S1).

## Motif groups to track colonizers' intra-guild indirect interactions through assembly

We developed four motif groups (see Fig. 3) to track how colonizers' intra-guild indirect interactions change throughout the transient period of their establishment (via species extinctions and subsequent establishments) to produce core motifs. As a result, these motif groups are not meant as a metric to characterize static pollination network structures. The motif groups have the following characteristics. First,

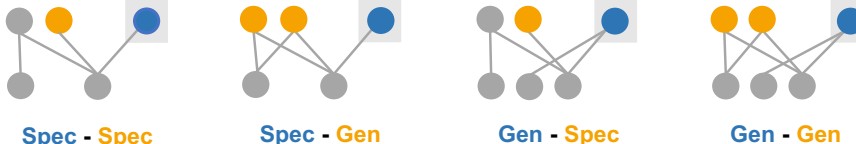

**Fig. 3 | Schematic representations of each motif group.** Each motif characterizes the niche breadth type (specialist or generalist) of the focal colonizer (blue) and the niche breadth type of intra-guild indirect partners (orange) as follows: (1) "Spec-Spec", specialist colonizer that interacts indirectly with *at least* one specialist in its guild; (2) "Spec-Gen", specialist colonizer that interacts indirectly with only generalists; (3) "Gen-Spec", generalist colonizer that interacts indirectly with at least

one specialist; (4) "Gen-Gen", generalist colonizer that interacts indirectly with only generalists. These motifs are used to answer guiding questions corresponding with each stage of core motif development (Fig. 1) which is illustrated for plants in Fig. 6 and pollinators in Fig. 8. Nodes in the same row represent species of the same guild (pollinators or plants).

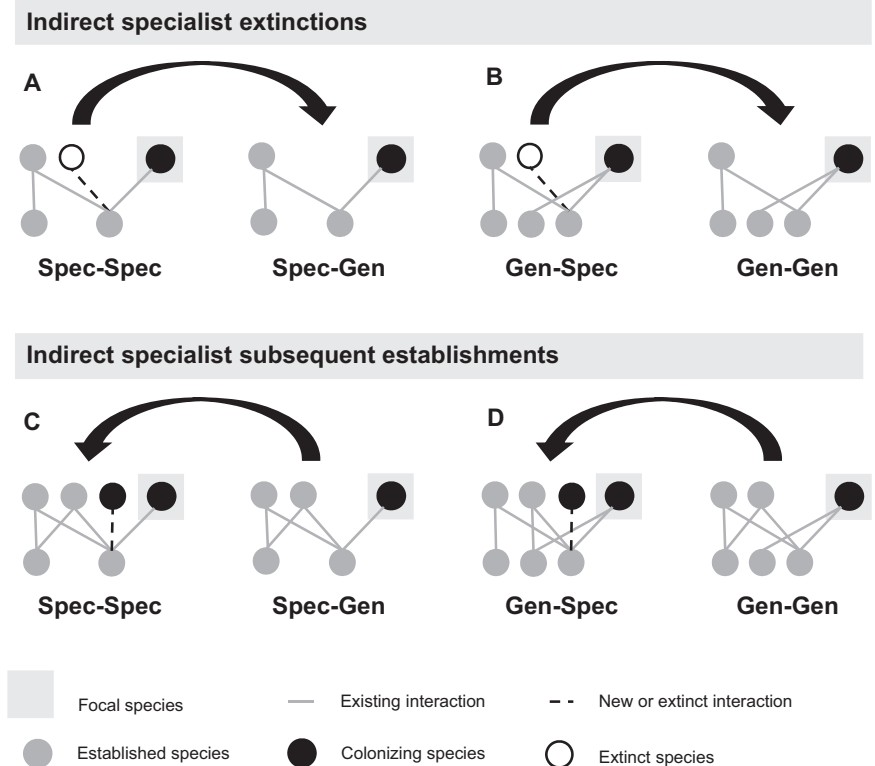

**Fig. 4 | Motif transformations identify intra-guild indirect specialist extinctions and subsequent establishments.** When all intra-guild indirect specialists are excluded by the focal colonizer, motif groups "Spec-Spec" and "Gen-Spec" are transformed to "Spec-Gen" and "Gen-Gen", respectively (**A**, **B**). When an intra-guild

indirect specialist colonizer establishes on the same mutualistic partner as the focal colonizer during the subsequent colonization event, motif groups "Spec-Gen" and "Gen-Gen" are transformed to "Spec-Spec" and "Gen-Spec" (**C**, **D**).

they are oriented by a focal colonizer species to examine the colonizer's direct and indirect interactions. Second, they emphasize the presence of specialists − species with one interaction − which distinguish motif dynamics by providing exclusive mutualistic resources and high-quality pollination services in the case of plants and pollinators, respectively. In addition, specialists in both guilds are the most vulnerable to disturbance due to inflexible niches. Third, a colonizer can belong to only one motif group at a time. As a result, we can analyze when colonizers' interactions transform from one motif group to another to reveal whether specialist extinctions or subsequent specialist establishments are taking place. For instance, if a specialist colonizer belonging to motif group "Spec-Spec" transformed to motif group "Spec-Gen", this would indicate that all intra-guild indirect specialists went extinct (see Methods, Fig. 4A). For every established colonizer, given they have and retain indirect interactions, we recorded their motif group at three times: the moment of their arrival, after

the extinctions they produce, and after the subsequent colonization event (corresponding to the guiding questions in Fig. 1). We then identified the common trends across all colonizers in each guild which produce the network's core motifs.

## Core motifs produced by plant species in the assembly model with adaptive foraging

Among colonizing plant species in the assembly model with adaptive foraging, specialists who share their one pollinator species with only generalist plants ("Spec-Gen") establish at the highest rate (24% of the 7227 attempted colonizers in the "Spec-Gen" motif group, Table S2; Figs. 5A and 6A). Indirect generalist plants offer fewer floral rewards (because they are depleted by other pollinators) than the specialist colonizer, causing the pollinator species to redirect its foraging effort from the generalist plants to the specialist colonizer. This competition between specialist colonizers and indirect generalist plants addresses

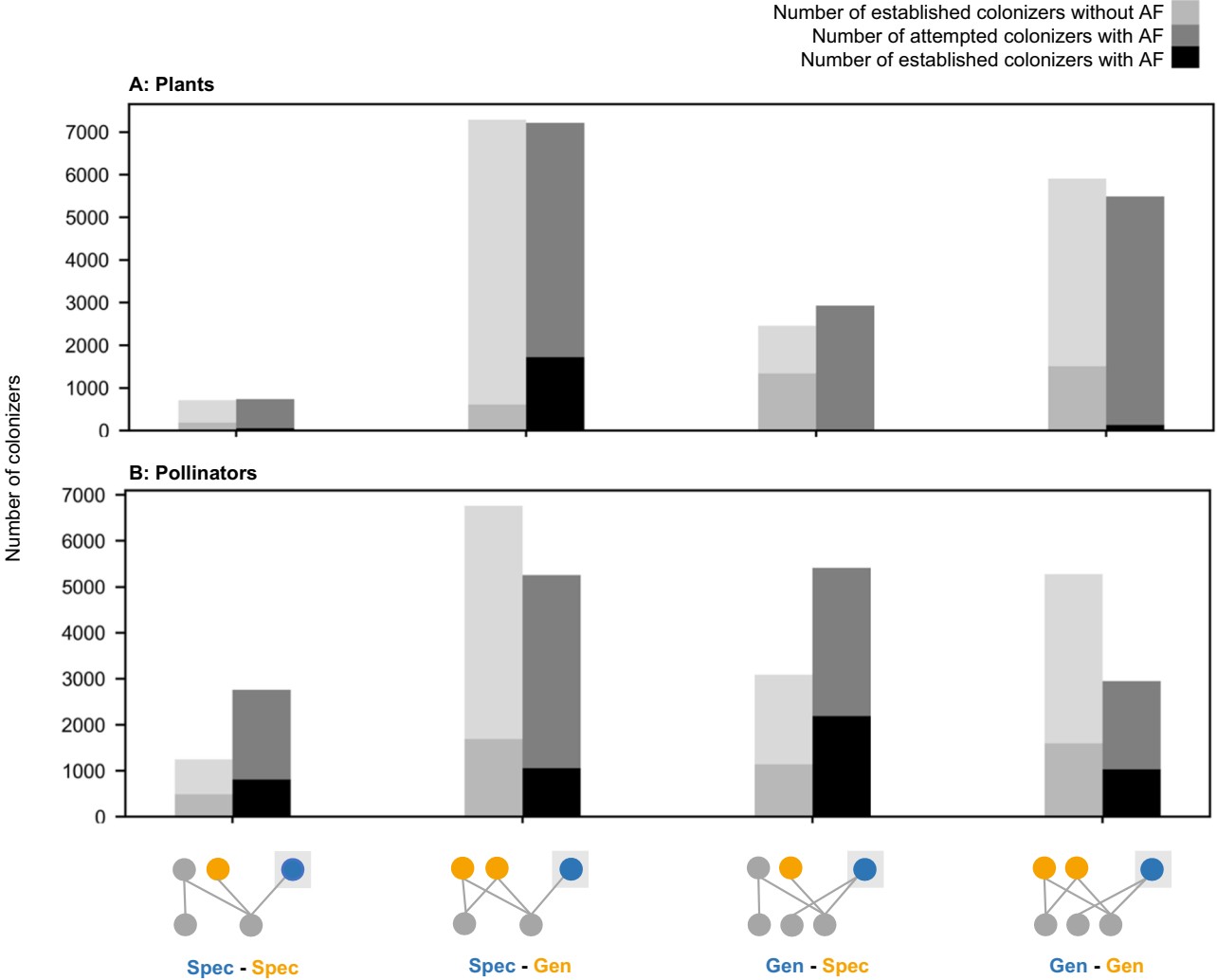

**Fig. 5 | Number of attempted and established colonizers across motif groups.** Here we consider species' motif groups at the moment they arrive to the network before undergoing any transformations due to specialist extinctions or subsequent establishments. Panels **A** and **B** show the number of attempted (light gray) and established (medium gray) colonizers in the assembly model without adaptive foraging (AF) as well as the number of attempted (dark gray) and established (black) colonizers in the assembly model with AF for plants and pollinators, respectively. The number of attempted pollinator colonizers in each motif group varies between the assembly models with and without AF due to network-level structure. Networks assembled from the model without adaptive foraging are significantly more connected and less nested (Fig. 2B, C) meaning those networks include fewer specialist pollinators connected to generalist hubs. As a result, attempted pollinator colonizers are less likely to have indirect interactions with specialists ("Spec-Spec" and "Gen-Spec") than those in the assembly model with adaptive foraging. The motif distribution of attempted colonizers does not vary significantly between early, middle, and late-stage colonizers. However, early colonizers establish at a higher rate than later colonizers (Fig S2).

our first question: how do intra-guild indirect interactions affect colonizer establishment? The specialist colonizer does not share its pollinator species with other specialist plants, so it cannot competitively exclude them (Fig. 6B) which addresses our second question: do colonizers competitively exclude intra-guild indirect specialists? Subsequent specialist colonizing plants rarely get established in this motif (3% of 1780 established colonizers in the "Spec-Gen" group following extinctions, Table S3 and Fig. 6C) as they cannot attract visits of the pollinator species foraging on the focal specialist colonizer which addresses our third question: do colonizers facilitate the establishment of subsequent colonizing intra-guild indirect specialists? Therefore, this motif ("Spec-Gen", i.e., specialist plant indirectly interacting with only generalist plants) is the most frequently produced among plants in the assembly process (52.4% of 1922 established colonizers across all motif groups, Table S3; Figs. 6D and 7A) which addressed our fourth question: what are the core motifs characterizing assembling networks?

## Core motifs produced by plant species in the assembly model without adaptive foraging

We see different dynamics for colonizing plant species in the assembly model without adaptive foraging. Generalist colonizers who share pollinators with at least one specialist plant species ("Gen-Spec") establish at the highest rate (54% of the 2460 attempted colonizers in the "Gen-Spec" motif group, Table S2; Figs. 5A and 6E). When pollinators are fixed foragers (per-capita they visit each plant species in their diet with equal foraging effort regardless of the distribution of floral rewards), colonizing plant species require many visits by abundant pollinators to get established. Pollinators visiting specialist plants in this model still benefit from exclusive rewards and become the most abundant. Therefore, generalist colonizers sharing pollinators with specialist plants receive the greatest quantity of visits overall. This shared pollinator abundance mediates indirect facilitation by a specialist plant to the generalist colonizer which addresses our first question: how do intra-guild indirect interactions affect colonizer

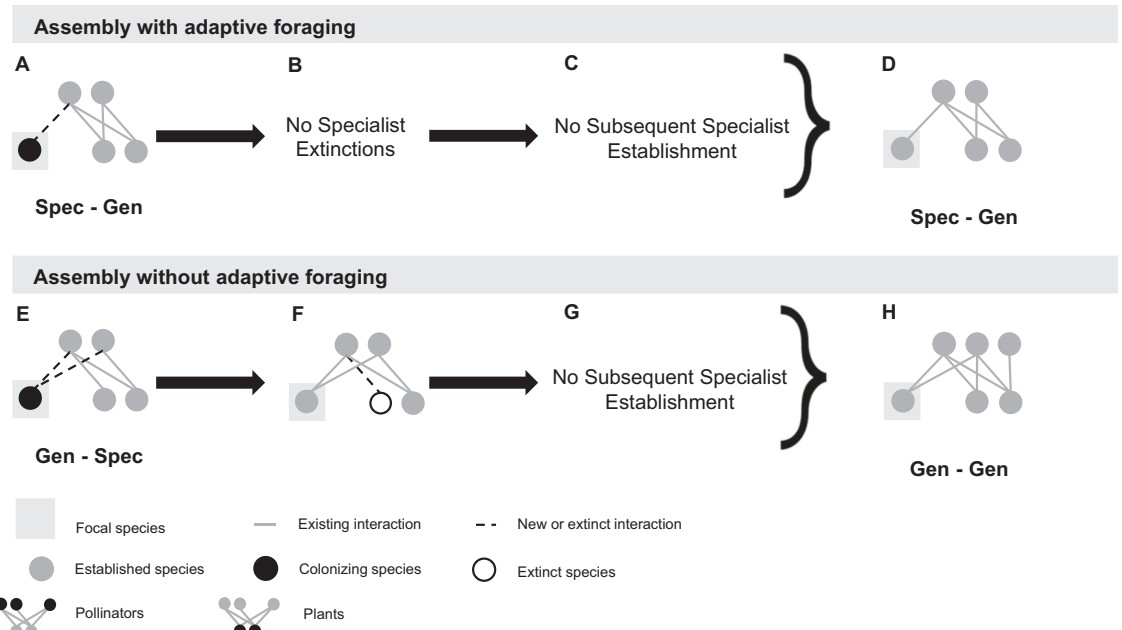

**Fig. 6 | Core motif development from colonizing plant species.** These diagrams represent the most common pathways producing the core motif group, however other pathways and motif groups were observed (see Fig. 7A and Table S3). With adaptive foraging, colonizers categorized as "Spec-Gen" established at the highest rate (**A**). Following establishment, intra-guild indirect specialists never went extinct (**B**, because there are no intra-guild indirect specialist in this motif) and were rarely established in the subsequent colonization event (**C**). Therefore the core motif produced by plants in the assembly model with adaptive foraging is "Spec-Gen" (**D**). Without adaptive foraging, colonizers categorized as "Gen-Spec" established at the highest rate (**E**). Following establishment, however, intra-guild indirect specialists frequently went extinct (**F**) and were rarely established in the subsequent colonization event (**G**). As a result, the core motif produced by plants in the assembly model without adaptive foraging is "Gen-Gen" (**H**).

establishment? However, as the generalist colonizer species grows in abundance, it frequently excludes indirect specialist plants (55% of the 1339 established colonizers in this motif group, Table S3 and Fig. 6F) thereby transforming motif group "Gen-Spec" into motif group "Gen-Gen". This addresses our second question: do colonizers competitively exclude intra-guild indirect specialists? Subsequent specialist colonizers who share their pollinator with the focal generalist colonizer rarely establish because they cannot secure enough visits (4% of the 2189 established colonizers in the "Gen-Gen" motif group following extinctions, Table S3 and Fig. 6G) which addresses our third question: do colonizers facilitate the establishment of subsequent colonizing intra-guild indirect specialists? As a result, the core motif produced by plant species in the assembly model without adaptive foraging is "Gen-Gen" (67% of 3645 established colonizers across all motif groups, Table S3; and Figs. 6H and 7A). This addresses our fourth question: what are the core motifs characterizing assembling networks?

### Core motifs produced by pollinator species in the assembly model with adaptive foraging

In the assembly model with adaptive foraging, colonizing pollinators that establish at the highest rate are generalists that indirectly interact with at least one specialist ("Gen-Spec"; 40% of 5415 attempted colonizers in the "Gen-Spec" motif group, Table S2; Figs. 5B and 8A). These colonizers are most successful because generalist pollinators can outcompete specialists for shared floral resources for two reasons. First, generalist pollinators have more floral resources available to them which makes them more abundant. Second, only generalist pollinators can adaptively forage as specialists have only one interaction and, therefore, no other options to reassign their effort. This competition between the generalist colonizer and indirect specialist pollinators addresses our first question: how do intra-guild indirect interactions affect colonizer establishment? However, generalist colonizers usually do not exclude indirect specialists (1% of 2189

established colonizers in the "Gen-Spec" motif group, Table S3 and Fig. 8B). Instead, adaptive foraging enables the generalist colonizer to quantitatively partition their niche which allows specialists to coexist on shared mutualistic resources which addresses our second question: do colonizers competitively exclude intra-guild indirect specialists? We cannot detect whether subsequent specialist colonizers get established in this motif because it would not transform the motif categorization. However, we know that specialists are less successful at establishing than generalists overall because they cannot compete for shared floral resources (Table S2 and Figs. 5B and 8C) which addresses our third question: do colonizers facilitate the establishment of subsequent colonizing intra-guild indirect specialists? Therefore, the core motif produced by pollinator species in the assembly model with adaptive foraging is "Gen-Spec" (i.e., generalist pollinators indirectly interacting with at least one specialist pollinator; 54.9% of 5091 established colonizers across all motif groups, Table S3 and Figs. 8D and 7B) which addresses our fourth question: what are the core motifs characterizing assembling networks?

### Core motifs produced by pollinator species in the assembly model without adaptive foraging

In the assembly model without adaptive foraging, specialist and generalist colonizing pollinators sharing floral resources with at least one specialist ("Spec-Spec" and "Gen-Spec") establish at the highest rate (39% of 1251 attempted colonizers in the "Spec-Spec" motif group and 37% of 3087 attempted colonizers in the "Gen-Spec" motif group, Table S2 and Figs. 5B and 8E). However, because there are relatively few attempted colonizers in the "Spec-Spec" motif group we will focus on the "Gen-Spec" motif group. Generalist pollinators outcompete specialists for shared floral resources by being more abundant which addresses our first question: how do intra-guild indirect interactions affect colonizer establishment? Without adaptive foraging, the competitive pressure exerted by generalist

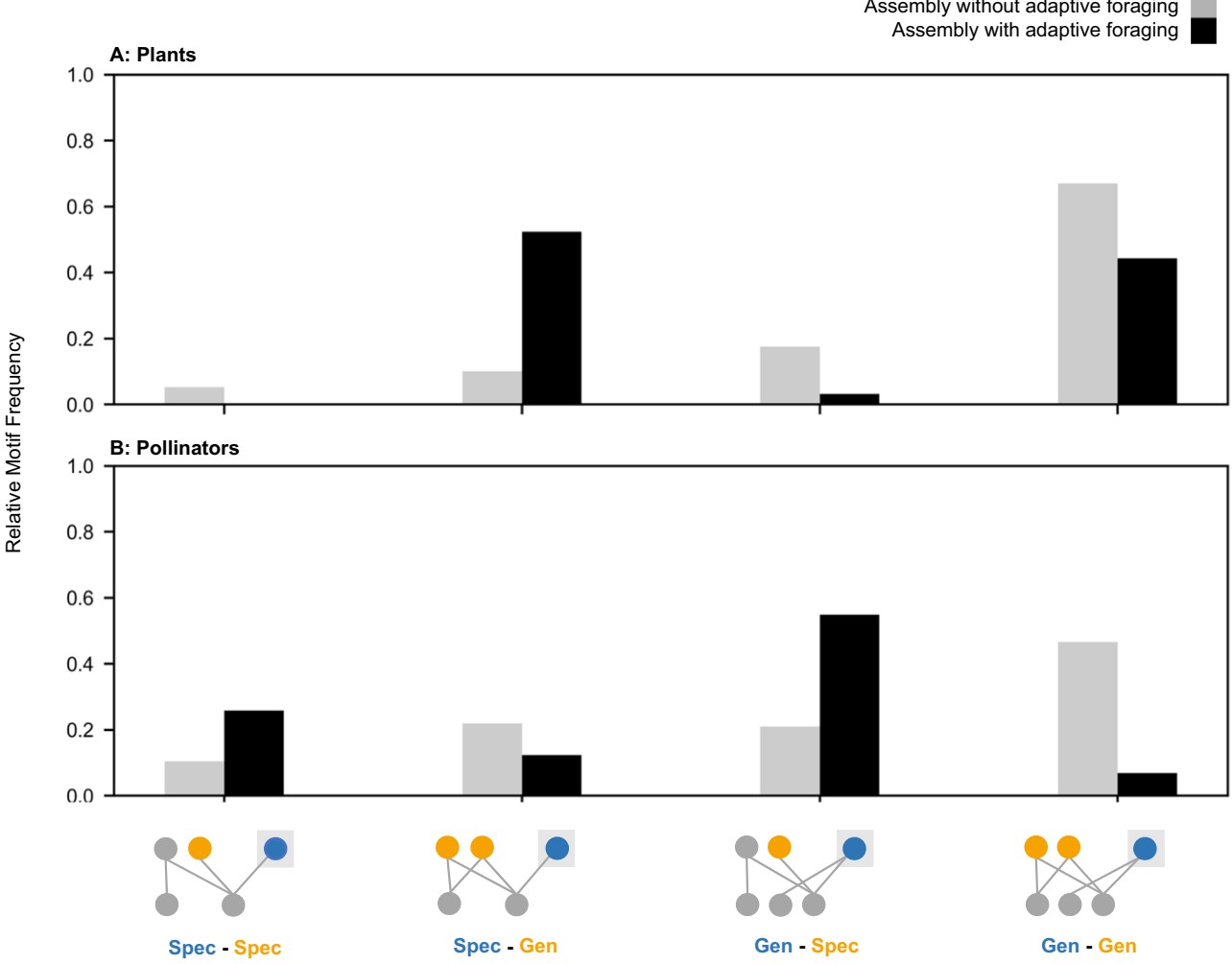

**Fig. 7 | Relative occurrence of each motif group following specialist extinctions and subsequent establishments.** Here we consider colonizer's motif groups following transitions due to specialist extinctions and subsequent establishment. Panels **A** and **B** show the occurrence of each motif group in the assembly model without adaptive foraging (AF, gray) and the assembly model with AF (black) for plants and pollinators, respectively. Among networks assembled with adaptive foraging, the motif "Spec-Gen" is the most common for plant species (**A**), while "Gen-Spec" is the most common for pollinator species (**B**). Among networks assembled without adaptive foraging, "Gen-Gen" is the most common for both plant (**A**) and pollinator species (**B**). The frequency of each motif group in each model corresponding with this graph can be found in the column "Motif Frequency after Subsequent Establishment" in Table S3.

colonizers often leads to the extinction of indirect specialists (24% of 1142 established colonizers in the "Gen-Spec" motif group, Table S3 and Fig. 8F) which transforms motif group "Gen-Spec" to "Gen-Gen". This addresses our second question: do colonizers competitively exclude intra-guild indirect specialists? Again, subsequent specialist colonizers sharing their plant species with the focal generalist colonizer rarely establish because they cannot compete for floral resources (9% of 1718 established colonizers in the "Gen-Gen" motif group after extinctions, Table S3 and Fig. 8G) which addresses our third question: do colonizers facilitate the establishment of subsequent colonizing intra-guild indirect specialists? The core motif produced by pollinator species in the assembly model without adaptive foraging is "Gen-Gen" (46.6% of 4915 established colonizers across all motif groups, Table S3 and Figs. 8H and 7B). Because fewer specialist pollinators are established in the network, "Spec-Gen" and "Gen-Gen" become more common among attempted colonizing pollinators (6761 and 5277, Table S2 and Fig. 5B) relative to the assembly model with adaptive foraging (5254 and 2960, Table S2 and Fig. 5B). This contributes to the dominance of motif group "Gen-Gen" overall. Together, this addresses our fourth question: what are the core motifs characterizing assembling networks?

## Discussion

Our results offer a new perspective on how complex networks assemble through the transient dynamics of colonizer establishment. Moving forward, we focus only on the results with adaptive foraging because empirical evidence shows that pollinators behaviorally increase their foraging effort on plant species in their diet with the most floral rewards available (reviewed in ref. 55) and because the networks assembled by our model with adaptive foraging were the most similar to empirical networks (Fig. S1). We found that intra-guild indirect interactions with species of opposite niche identities ("Spec-Gen" for plants and "Gen-Spec" for pollinators in the assembly model with adaptive foraging, Fig. 5 and Table S2) was advantageous for colonizer establishment because of the competitive pressure colonizers can leverage. As evidenced by the assembly model without adaptive foraging, these motifs are ultimately unstable resulting in the competitive exclusion of specialists. However, they persist in the assembly model with adaptive foraging because adaptive foraging enables pollinators to quantitatively partition their niches[43,55]. This coexistence mechanism produces nested networks because generalist and specialist species must share mutualistic resources for networks to be nested. Further, by preserving the persistence of specialists, this

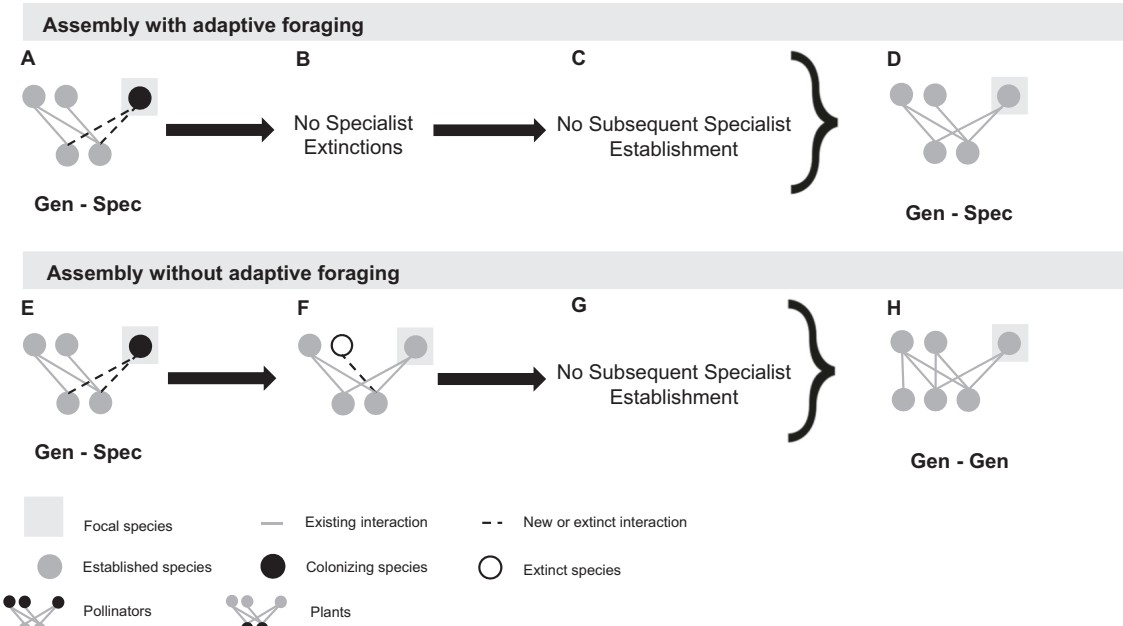

**Fig. 8 | Core motif development from colonizing pollinator species.** These diagrams represent the most common pathways producing the core motif group, however other pathways and motif groups were observed (Fig. 7B and Table S3). With adaptive foraging, colonizers categorized as "Gen-Spec" established most frequently (**A**). Following establishment, intra-guild indirect specialists rarely went extinct (**B**) and if they were established during the subsequent colonization event it would not change the motif categorization (**C**). Therefore, the core motif produced by pollinators in the assembly model with adaptive foraging is "Gen-Spec" (**D**). Without adaptive foraging, colonizers categorized as "Gen-Spec" also established at the highest rate (**E**). Following establishment, however, intra-guild indirect specialists frequently went extinct (**F**) and were rarely established in the subsequent colonization event (**G**). As a result, the core motif produced by pollinators in the assembly model without adaptive foraging is "Gen-Gen" (**H**).

produces networks that are richer and less connected which is also consistent with empirical networks (Fig. S1). These findings support theory that mutualisms can mediate species coexistence through niche differences enabled by adaptive foraging[57], and that mechanisms of coexistence between indirectly connected species determine the structure and stability of novel species assemblages[7]. In the following paragraphs, we answer each of our guiding questions (Fig. 1) and contextualize our results with prior empirical and theoretical literature.

### How do intra-guild indirect interactions affect colonizer establishment?

We found that colonizers establish when they have a competitive advantage over species of the same guild that share their mutualistic resources. For pollinators, successful colonizers were generalists who are the most abundant and have the greatest niche flexibility. This finding is consistent with empirical literature on the two most well-known invasive pollinators in the United States: the European honeybee (*Apis mellifera*) and bumblebee species (*Bombus* spp.) such as *Bombus terrestris*. These species are highly generalized in the communities they invade and suppress native pollinator activity due to exploitative competition[58–65]. In terms of colonizers' niche flexibility enabled by adaptive foraging, empirical literature also supports that pollinator species with high niche flexibility are highly successful colonizers[66].

For plants, we found that the most competitive species were specialists because they offered the most exclusive floral rewards. This is supported by empirical studies showing that invasive plants have abundant rewards and reduce visitation to native plants[54,67–70]. Other empirical studies also show that invasive plants are more specialized than native plants on the most generalist pollinators[71–73]. In contrast, some studies have found invasive plants to be highly generalist;[30,74,75] this discrepancy could be due to comparing early and late stages of invasion. Invasive species beginning at low abundances are often only visited by the most abundant pollinator species but accumulate additional interactions with less abundant pollinators as they become established[76].

### Do colonizers competitively exclude intra-guild indirect specialists?

We found that colonizers generally do not exclude indirect specialist competitors due to niche partitioning enabled by adaptive foraging. This finding is consistent with coexistence theory that species competing for mutualistic commodities can coexist through niche and fitness differences[57,77]. While many empirical studies attribute niche partitioning to morphological or phenological traits[78–80], our results suggest that pollinators further partition their niches through adaptive foraging in response to resource abundance and interspecific competition[81]. The empirical evidence relating to this trend is mixed. For instance, despite both pollinators being highly generalist, *A. mellifera* tends to occupy distinct ecological niches from native pollinators and does not exclude them, while *Bombus* spp. does exclude native pollinators[82,83].

### Do colonizers facilitate the establishment of subsequent colonizing intra-guild indirect specialists?

We found that subsequent specialist colonizers of the same guild did not establish within the motif of the focal colonizer due to competitive pressure. This is supported by empirical evidence that invasive plants[84,85] tend to fulfill distinct ecological niches in networks. However, there also exists evidence that invasive species can facilitate one another[85], a process known as invasion meltdown[86]. Invasion meltdown among species within a guild through shared mutualistic resources is relatively understudied;[21] only a few empirical examples have been documented[29,75]. Many studies, however, have found that species indirectly facilitate one another through suppressing a shared

competitor[21,87,88]. Detecting this would require increasing the size and complexity of motifs, a potential avenue for future research.

**What are the core motifs characterizing assembling networks?**
We found that adaptive foraging dynamics produce core motifs in which specialists and generalists coexist on a shared mutualistic resource which produces nested species interactions, a feature commonly observed in mutualistic networks. While we have considered the establishment of colonizing plants and pollinators separately, the core "Spec-Gen" and "Gen-Spec" motifs for plants and pollinators, respectively, can also emerge together. Specialist plant colonizers in motif group "Spec-Gen" can provide exclusive resources to generalist pollinator colonizers in motif group "Gen-Spec" who can in turn perform high-quality visits by quantitatively specializing on those plants. The "mega-motif" combining plants' "Spec-Gen" motifs and pollinators' "Gen-Spec" motifs is likely to emerge given the positive direct effects between colonizers. Moreover, this "mega-motif" is nested in both guilds which magnifies the emergence of nested networks.

We are aware of a couple other studies that have explained the emergence of nestedness through the dynamics of indirectly connected species[38]. In the model developed by Bastolla et al.[38], they found that when direct competition is weak, indirect effects are exclusively facilitative and colonizing species require this indirect facilitation to overcome competition for non-mutualistic resources and establish[38,89]. Therefore, specialist colonizers can only establish and persist if they interact with generalists through whom they have many indirect connections; this produces nestedness. However, when direct competition exceeds a critical value, indirect effects become competitive leading to the competitive exclusion of specialists by indirect generalists[89]. Duchenne et al.[28]. expanded on Bastolla's model[38] to investigate how network structure arising from morphology and phenology influences the nature of indirect interactions. They found that morphology increases indirect competition while phenology increases indirect facilitation which produces more diverse and nested networks in which intra-guild indirect generalists and specialists can coexist. While our assembly model is dominated by indirect competition, we observed the emergence of nested networks because adaptive foraging alleviates the strength of indirect competition to maintain species coexistence. Importantly, however, adaptive foraging does not reverse indirect effects to make them facilitative. Therefore, our results suggest that destabilizing nested interactions arise from colonizers leveraging indirect competition, and these interactions persist due to adaptive foraging.

Our motif analysis is limited by only considering indirect interactions through a path length of two despite the fact that previous studies have found evidence of significant indirect effects through longer paths[24–27]. Future studies could use mathematical approaches to identify key indirect effects and design motifs around those to investigate their influence on network structure. Another limitation of our analysis is that the occurrence of each motif group among attempted colonizers is biased by the network structure. For instance, because the "Spec-Spec" and "Gen-Spec" motif groups are defined by species having *at least* one indirect interaction with a specialist, in larger networks where species have more indirect interactions, it is more likely that at least one of those indirect partners will be a specialist. Similarly, "Spec-Spec" and "Gen-Spec" are more common in less connected and more nested networks because there are more specialists connected to generalist hubs, such as in the networks from our assembly model with adaptive foraging. This suggests feedback in assembly: colonizer establishment produces network-level structure and over time network-level structure influences colonizer establishment. Therefore, tracking network-level structural development is necessary to comprehensively characterize assembly dynamics.

Our theoretical approach overcomes temporal and spatial limitations in empirical research which would require observed network data at the scale of 150 years sampled at a fine temporal resolution (i.e., at least every 3 years). However, future empirical studies could test the following key results: (1) colonizers (species arriving via biological invasions, phenological shifts, or range shifts) contribute fundamentally competitive motifs to assembling plant-pollinator networks, and (2) pollinator niche flexibility allows those motifs to persist over time. The first result would require testing whether the colonizer increases (facilitative) or decreases (competitive) the population growth rate of resident species in their guild with whom they share mutualistic partners, or, as a proxy, the abundance of mutualistic resources/services available to those species. The latter combines the quantity of resources/services the colonizer consumes and the quantity they generate indirectly via increasing the population growth rate of shared mutualistic partners. To evaluate this empirically between colonizing and resident pollinator species, one could measure the quantity of nectar consumed and pollination services provided to shared plants. The effect of pollination services on plant demography would then need to be empirically measured to determine whether the colonizer contributed a net positive (facilitative) or negative (competitive) effect on the nectar available to resident pollinators. The second result would require empirically measuring whether resident pollinators redistribute their foraging effort among plant species in their niche in response to the colonizer's establishment. This could be done by evaluating whether resident pollinators have wider (more generalized) realized niche breadths than they did prior to colonization and/or whether the distribution of foraging efforts among species in their niche has shifted. Finding evidence for these two results would allow empiricists predict the trajectory of assembling network structure, dynamics, and stability.

In conclusion, our work advances the field of community assembly by unveiling how the transient dynamics of colonizer establishment influence network structural development over long timescales. Specifically, we found that colonizers leverage competition with species in their guild for shared mutualistic resources to establish and that adaptive foraging maintains the persistence of inferior competitors. This produces motifs in which intra-guild indirect species possess variable niche breadths which are the building blocks for nested networks.

## Methods
### Population dynamics
We used Valdovinos et al.'s model[53] to simulate the population dynamics of each plant ($P_i$, Eq. 1, see below) and pollinator ($A_j$, Eq. 2) species of the network, as well as the dynamics of floral rewards ($R_i$, Eq. 3) of each plant species, and the foraging effort ($\alpha_{ij}$, Eq. 4) that each pollinator species (per-capita) assigns to each plant species, as follows:

$$\frac{dP_i}{dt} = \gamma_i \sum_{j \in A_i} e_{ij} v_{ij} \sigma_{ij} - P_i \mu_i^P \qquad (1)$$

$$\frac{dA_j}{dt} = A_j \sum_{i \in P_j} c_{ij} f(R_i) - A_j \mu_j^A \qquad (2)$$

$$\frac{dR_i}{dt} = \beta_i P_i - \phi_i R_i - \sum_{j \in A_i} A_j f(R_i) \qquad (3)$$

$$\frac{d\alpha_{ij}}{dt} = G_j \alpha_{ij} \left( \frac{c_{ij} f(R_i)}{\alpha_{ij}} - \sum_{k \in P_j} c_{kj} f(R_k) \right) \qquad (4)$$

This model assumes plant and pollinator species are obligate mutualists. The population growth of plant species $i$ (Eq. 1) is governed by the quantity ($v_{ij}$) and quality ($\sigma_{ij}$) of visits it receives from each of its

pollinator species, the expected number of seeds produced by a pollination event ($e_{ij}$), the fraction of those seeds that recruit to adults ($\gamma_i$), and mortality loss ($\mu_i^P$). Visit quantity and quality are defined as:

$$v_{ij} = A_j P_i \tau_j \alpha_{ij} \qquad (5)$$

$$\sigma_{ij} = \frac{\epsilon_i v_{ij}}{\sum_{k \in P_j} \epsilon_k v_{kj}} \qquad (6)$$

where $\tau_j$ is the visitation efficiency pollinator species $j$. Visit quality (Eq. 6) is the amount of conspecific pollen that pollinator species $j$ carries of plant species $i$ relative to the total amount of pollen the pollinator carries from all the plant species it visits. This amount is proportional to how much pollen is collected from each plant species during a single visit ($\epsilon_i$). The fraction of seeds that recruit to adults of plant species $i$ ($\gamma_i$ in Eq. 1) is defined as follows:

$$\gamma_i = g_i \left(1 - \sum_{l \neq i \in P_j} u_l P_l - w_i P_i\right) \qquad (7)$$

where $g_i$ is the maximum fraction of seeds that recruit to adults, $u_l$ is the interspecific competition for non-mutualistic resources, and $w_i$ is the intraspecific competition for non-mutualistic resources.

The population growth of pollinator species $j$ (Eq. 2) is governed by the pollinators' rewards conversion efficiency ($c_{ij}$), mortality loss ($\mu_j^A$), and rewards consumption ($f(R_i)$), which is defined as:

$$f(R_i) = \alpha_{ij} \tau_j b_{ij} R_i \qquad (8)$$

where $b_{ij}$ is the pollinator species $j$'s extraction efficiency of rewards of plant species $i$ in each visit. This consumption scales linearly with respect to floral rewards abundance ($R_i$). For simplicity, we assumed a Type I functional response because the dynamics of plant rewards are constrained by a saturating production rate ($\phi_i$) which, in turn, saturates pollinator consumption.

The growth of plant $i$'s floral rewards (Eq. 3) is governed by its production rate ($\beta_i$), saturation rate ($\phi_i$), and consumption from all of the pollinator species that visit plant species $i$. Finally, the foraging effort pollinator $j$ allocates to plant $i$ (Eq. 4) is governed by the rate of adaptive foraging ($G_j$), pollinator resource conversion efficiency ($c_{ij}$), resource consumption from plant $i$ per unit effort, and average resource consumption across all the plant species pollinator $j$ visits (Eq. 4). Foraging effort takes values between 0 and 1, and the sum of $\alpha_{ij}$ over all plant species that pollinator species $j$ visits sums to one.

Each colonizer is assigned values for each parameter by sampling a uniform distribution of a given mean and variance (Table S1). Several studies have previously analyzed this model and its sensitivity to parameter values; however, this is the first time it has been used in the context of community assembly. To ensure species turnover in our assembly model, we increased interspecific species competition by modifying key parameter values[77,90,91]. For pollinators, we increased the variance of visitation efficiency ($\tau_j$) and for plants we increased the mean and variance of pollen production ($\beta_i$), amount of pollen collected during a single visit ($\epsilon_i$), and interspecific competition for non-mutualistic resources ($u_l$) and decreased the mean and variance for intraspecific competition for non-mutualistic resources ($w_i$).

## Simulation design

We developed two dynamic assembly models, one including and one excluding adaptive foraging to serve as a null model. Assembly begins with an empty network in which three plant and three pollinator species attempt to colonize the network every 2000 timesteps for a total of 50 colonization events. Given that the per-capita mortality rate of all

pollinators in our model, $\mu_j^A$, is equal to 0.001, a generation (the time it takes a population to double, Eq. 9) is about 693.15 timesteps.

$$T_j = \frac{\ln 2}{\mu_j^A} \qquad (9)$$

Therefore, the $10^5$ timesteps that each of our simulations take corresponds to about 144 generations of pollinators. Given that pollinators commonly reproduce annually, each simulation roughly spans 144 years and colonization events occur every 3 years. Colonizing species were introduced at abundances set to the extinction threshold given that in nature species often arrive in new habitats with very few individuals. For each model, a suite of simulations was performed in which we incrementally varied the values of two parameters corresponding to the probability that colonizing plants or pollinators will be specialist (11 probabilities for each guild ranging from 0 to 1, $11 \times 11 = 121$ simulations per model). A specialist species has only one interaction upon introduction while the degree of a generalist species is drawn randomly from a uniform distribution ranging from two to the maximum number of partners in the network. Among the attempted colonizers, only a proportion successfully establish in the network and between each colonization event the model reaches a steady state. Following establishment, colonizers' degrees are unrestricted. As a result, many species gain interactions the longer they persist in the network, and generalist species added to larger networks are likely to have higher degrees.

## Motifs

Four motif groups were developed to characterize the indirect interactions of colonizing species. They are distinguished by whether the colonizing species is generalist or specialist, and whether they interact indirectly with only generalists or at least one specialist; diagrams of each can be seen in Fig. 3. The only colonizers excluded from these four motif groups are those that directly interact with only specialists and therefore form distinct modules (disconnected from the rest of the network) without any indirect interactions. In our simulations, this occurred either because colonizers were only attached to specialists when they were introduced or because all species indirectly connected to the colonizer went extinct during colonizer establishment. Both situations occurred rarely, and distinct modules always reconnected to the network during subsequent colonization events. Therefore, by excluding these colonizers we did not lose significant information about the assembling structure of plant-pollinator networks. In addition, we excluded from the motif analysis the first nine colonizers of each guild populating the network because the network was too small to analyze.

Motif development (Fig. 1) was evaluated only for colonizers that successfully establish in the network. In the context of this study, those are the species that survive longer than 4000 timesteps. For each established colonizer, their motif group was evaluated at the time of their introduction, 2000 timesteps later when perturbed species were extinct, and 4000 timesteps later when the subsequent colonizers had established. At each stage, we tracked whether colonizers' interactions were transformed from one motif group to another to determine where specialist extinctions and subsequent establishments took place. If all intra-guild indirect specialists are excluded by the focal colonizer establishment, motif groups "Spec-Spec" and "Gen-Spec" will be transformed to "Spec-Gen" and "Gen-Gen", respectively (Fig. 4A, B). If during the subsequent colonization event intra-guild indirect specialist colonizers establish on the same mutualistic partner as the focal colonizer, motif groups "Spec-Gen" and "Gen-Gen" will be transformed to "Spec-Spec" and "Gen-Spec" (Fig. 4C, D). Motif group transformations can also result from extinctions and subsequent establishments of species in the opposite guild. For instance, focal generalists can transition to specialists due to extinctions of direct partners and focal

specialists can transition to generalists due to the subsequent establishment of direct partners. However, we do not track these transformations because they are not the focus of this study.

## Network structure

At the end of each simulation (timestep $10^5$) we measured the network's binary interaction structure (121 networks per assembly model). This included plant degree distribution, pollinator degree distribution, richness, connectance, pollinator: plant ratio, and nestedness. Nestedness was calculated for each network with the NODFc metric to compare across networks of varying size and connectance[55]. We performed one-sided Welch $t$-tests to statistically evaluate the hypotheses that networks assembled from the model with adaptive foraging are richer, less connected, have a higher pollinator:plant ratio, and are more nested than networks assembled from the model without adaptive foraging. The Welch tests were paired to compare networks populated by specialist plants and specialist pollinators at the same probability.

## Reporting summary

Further information on research design is available in the Nature Portfolio Reporting Summary linked to this article.

## Data availability

Simulated plant-pollinator network assembly data can be reproduced with the code and instructions available at the project's GitHub repository[92] v1.0.0: https://github.com/Valdovinos-Lab/Motif_Assembly. Empirical plant-pollinator network data obtained from the Web of Life: ecological networks database is available here: https://www.web-of-life.es/map.php?type=5.

## Code availability

Michael Egan and Fernanda Valdovinos developed a MATLAB toolbox called "PlantPollinator_Network_Builder.mltbx" to perform simulated assembly of plant-pollinator networks based on Fernanda Valdovinos' theoretical consumer-resource model of plant-pollinator interactions. Sabine Dritz additionally developed code to perform the motif analysis presented in this study. The toolbox, code, and instructions to reproduce the results of this study are available at the project's GitHub repository[92] v1.0.0: https://github.com/Valdovinos-Lab/Motif_Assembly.

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

## Acknowledgements

We thank Michael Egan for his assistance in programing the MATLAB package used to simulate network assembly with the Valdovinos et al. 2013 model. We also thank the Valdovinos Lab, particularly Kayla Hale, for providing feedback on the manuscript. This work was funded by NSF grant DEB-2129757.

## Author contributions

F.S.V. conceived and conceptualized the study and developed original assembly and population dynamic codes. S.D. and F.S.V. designed the study, developed the methods, performed the data analysis, and created figures. S.D. and R.A.N. wrote the first draft of the manuscript with important contributions of F.S.V. and contextualized the study with related empirical work. All authors contributed to the final version of the manuscript.

## Competing interests

The authors declare no competing interests.
