## [Peer Review File · Nature Communications]

The role of intra-guild indirect interactions in assembling plant-pollinator networksReviewers' Comments:

Reviewer #1:

Remarks to the Author:

Review of "The role of intra-guild indirect interactions in assembling plant-pollinator networks," by Dritz et al.

The authors investigate the role of intra-guild indirect interactions on plant-pollinator network assembly using motif analysis and a colonization model that incorporates adaptive foraging. They find that adaptive foraging promotes the coexistence of specialists and generalists within the same guild and that colonizers tend to form intra-guild indirect interactions with species of opposite niche breadth (specialist colonizers with generalist incumbents for plants and generalist colonizers with specialist incumbents for pollinators).

The manuscript is generally well written, especially the discussion, and the core ideas---motif analysis for studying indirect interactions and the effects of adaptive foraging on community assembly---are novel and interesting. My main critique is that the study is heavily theoretical/modelling/simulation-based and would really benefit from additional analyses involving empirical networks. I also have some other suggestions for improving the manuscript.

-- Tighter integration of empirical networks in analyses

The authors provide some comparison of their theoretical work to empirical networks (L255, L260, etc.), but these analyses are relatively limited and appear as more of an afterthought. While I like the overall modelling approach and find the theoretical results interesting, it will be important to see whether the findings translate to empirical networks. I appreciate that it is not possible to study the assembly of these empirical networks, but can the authors do something like provide a breakdown of motif frequencies in empirical networks? Right now, the comparison centers on three broad network properties (richness, connectance, and nestedness, see Fig. 6) and the results themselves are not especially clear and convincing and, moreover, do not provide any additional support to the main findings regarding intra-guild competition and higher-level interaction patterns.

-- Additional temporal analyses

The authors record motif groups in sets of three subsequent time steps (L124) but then group the motif data for all colonizers, regardless of when they established in a network (L127). It would be very interesting to see an explicit breakdown of the distribution of these three-motif sequences (i.e., similar to information presented in Table 1). It would also be cool to see if/how motif distributions change during the colonization trajectory, e.g., early vs. middle vs. late stages.

-- Remind the reader of the four questions

I like the four research questions (L76) but found it difficult to remember each question when they were referred to simply by Q1 etc. This was especially the case when reading the section "Core motifs produced by plant species" (L142). It would be helpful if sentences could be rewritten to remind readers of each question as the answers are presented.

-- Clarify what information is presented in Table 1 and Fig. 4

I did not find the information in Table 1 very easy to interpret. Is the establishment rate over all 121 simulations? Is such information worth presenting by model (W/ AF etc.) rather than broken down by plants vs. pollinators? Which rates are meant to sum to 1? The bottom line is that I did not find Table 1 very effective. Also, it wasn't clear to me if Fig. 4 (which I did find intuitive) presented the same/similar information to Table 1.

-- Consider specialists as having more than one interaction

I would be curious to know if it would be possible to perform the analysis with specialists defined by a diet breadth greater than one (L218)? Could it also work if specialists had a potential diet breadth greater than one but only realized one interaction at a time? In a similar vein, do the authors think results would change significantly if species were categorized into specialists, generalists, and super-generalists (with motifs reflecting this categorization)? Such an extension would help bring the work closer to empirical systems, in which diet breadths follow distributions that are much more complicated than just specialist having 1 interaction and generalists having > 1 interactions. Also, in simulations the number of interactions of a generalist is drawn from a uniform distribution (L388)---would results change if other distributions were used, particularly ones that more closely match empirical degree distributions?

-- Describe how results for plant-focused motifs fit with pollinator-focused motifs

Results for pollinators are by-and-large considered **separately** from results for plants. I would like to see a discussion on how the results regarding pollinators (e.g., generalist colonizers with specialist incumbents) and plants (e.g., specialist colonizers with generalist incumbents) might emerge **together** from a modelling/technical standpoint; this could go around L253.

Minor comments

-- Fig. 1. I don't know what a "passing" interaction is? Is there a better term that can be used instead? Maybe "transient" or "temporary"? Or perhaps add a definition in the caption.

-- L109. Add a short explanation of the "dynamic species pool" and how it works.

-- L122. It would be helpful to have an example of how "colonizers' interactions transform from one motif group to another." I had a hard time imaging what this would look like.

-- Fig. 2. I think the figure title would be clearer if it was, "...each motif group." Also, personally I would find "Spec" easier to read than "Spc".

-- Fig. 3. Because most plant-pollinator networks are drawn with plants as the lower guild and pollinators as the upper guild, it would be easier for me to follow if figures reflected this convention.

-- Fig. 6. I did not understand what was done based on the sentence (L272) beginning, "We performed a one-sided..." Consider expanding the technical description.

-- L327. I found the argument involving *A. mellifera* and *Bombus* spp. confusing since the example with *Bombus* spp. doesn't seem to support the general thesis of the paragraph.

Reviewer #2:
Remarks to the Author:
Cf. attached file

General comment

Dritz *et al.* developed a dynamic model to describe the assembly of mutualistic network, a surprisingly overlooked aspect of mutualistic networks. By including adaptive foraging in this article, they provided an innovative theoretical approach of the question, allowing them to account for the plasticity of interactions, that is often neglected. By combining this model with a motifs approach, they could track the impact of colonization on the network over the assembly process. This article is definitely a nice piece of science that provides new results about the dynamics of invasion in a network context. The introduction and discussion are well written and clear. I have no major comments about that manuscript. I have two main points of discussion I would like to raise: 1) about the use of motifs and possible bias and limits; 2) for the moment big parts of the methods are quite obscure and require a lot of energy from the reader to be understood because we need to read previous articles about the model, and the format (methods at the end) does not help. I detail a bit these points in the following paragraphs. Otherwise figures and results could be a bit clearer, but as I said above, these are no major issues and this article is really a worthy reading, thanks to the authors for that.

Authors focused on indirect effects that propagate through paths of length 2 only, so the shortest indirect effects possible, but this is never highlighted, while it would be nice to discuss this point. However, many studies have shown that in diverse networks like those studied by authors, an important part of indirect effects propagate through long-paths (Higashi & Nakajima 1995; Nakajima & Higashi 1995; Guimarães *et al.* 2017; Pires *et al.* 2020). The same studies using the Jacobian matrix of dynamical systems to study indirect effects over all these possible paths (Nakajima & Higashi 1995), that in contrast to the motif approach allow to integrate these effects and measure their strength, including abundances and interaction strengths to calculate propagation from species to species. Thus, I wonder why did authors prefer the motif approach relative to this method? About motifs analyses, also see my comment for lines 395-406, that is, I think, of importance.

The model is not understandable without referring to other papers, because parameters and variables are not described at all in the main Methods and partially in Appendix S2, which is, I think, a problem to understand the study. Some important equations are missing, for example the equation linking V_{ij} to α_{ij} and to species abundances. To summarise, in the absence of further explanations and descriptions the equations presented in Methods are useless. Nevertheless, once the reader has done the effort to read the previous paper (Valdovinos *et al.* 2013), this model is definitely a nice piece of work and this paper a nice piece of science. But more efforts are needed in explaining (again) the model in the present study.

Point-by-point comments

Line 38: I do not know this paper but by quickly reading through I do not see how ref. 20 (Bogdziewicz *et al.* 2018) is linked to this point. I might have missed something, but good to check if it is the right reference.

Lines 75-79: here and in the following parts, I started to have a problem of terminology to fully understand. In question 1 a colonizer is a species that attempts to colonize the system, regardless of the success of the colonization. But in following questions, a colonizer becomes a successful colonizer. I think it is important to distinguish both. When authors study motifs, do they present the result only for successful colonizer or all colonizers?

Edit: answer to that question is present lines 407-413, but since Methods are at the end might be good to find a way to clarify it in the main text.

Lines 78-79: at this point, it is *not* intuitive what is behind question 3 for me. I struggle to see where this question goes, which kind of mechanism it aims to explore (competitive exclusion? niche partitioning?) and what are the assumptions of authors.

Lines 111-113: without further explanation, this sentence is very enigmatic. It could be either developed here, or removed from here and developed in Methods.

Line 116: before entering in motif description, I really missed basic information on network size at the end of the process. So, it starts by 3X3 networks, and could end with 150X150 networks if all colonizing species survived and there is no extinction. Knowing the average network size at the end of simulation would be helpful to understand motifs analyses.

Edit: After continuing reading I found richness and connectance plots latter, but I really think it misses some description of the dynamics and end point of the simulation before describing the motifs part, otherwise it is really abstract. Authors could have a first plot at this stage that describe the dynamics of simulations and number of successful colonization and extinctions. Something in the spirit of what is below.

Lines 124-127: I should admit that for me it was not clear all long the article I had the feeling I had to guess when authors used the motifs count calculated at the moment of the colonization, after the extinctions, or after the subsequent colonization event. I think it could be labelled more clearly.

Fig. 2: I think that a bit of colours could help to understand better the figure.

For example:

Spc - Spc

Fig. 2 | Schematic representations of each motif. Each motif characterizes the niche breadth (specialist or generalist) of the focal colonizing species (specie highlighted by the box) and the niche breadth of species that share its mutualistic partner(s), as follows: 1) “Spc - Spc”, specialist colonizer (blue) that interacts indirectly with at least one specialist in its guild (orange): ...

Fig. 3: in C, it took me time to guess that authors mean “No subsequent specialist establishment”, as subsequent is missing.

It is a just a convention that can be changed but having the plants as the top guild of the bipartite network is not intuitive (they are often represented as the bottom guild, with pollinators as top guild). Since the paper is complex, I would stick to classic norms as much as possible to keep the readers focusing on important complexity.

Table 1: so this table correspond to motifs account at the moment of colonizer’s arrival? Could be good to be very clear about that.

Lines 285-288: Isn’t table 1 instead of Figure 4 that shows this result?

Lines 293-297: I find the wording “mutualisms mediate species coexistence” a bit weird. I would have said “adaptive foraging mediates species coexistence”, as without adaptive foraging mutualism does not maintain high coexistence.

Lines 346-350: About the emergence of nestedness through indirectly connected species it could be worthy to link this result to similar results found when introduced a phenological structure in mutualistic networks (Duchenne *et al.* 2021). Like adaptive foraging, phenological structure also protects specialist from extinctions through indirect effects, and thus promote nestedness and stable coexistence of a higher diversity (Duchenne *et al.* 2021).

Lines 376 – 380: I could not find the meaning and units of γ_i , σ_{ij} , V_{ij} .

Variable names are not described. Often for readability variables are represented with capital letters, while parameters with lowercase letters, I think following this convention would help to understand quickly the equations, that are here really hard to understand, especially without description.

Basic assumptions of the model (*e.g.* linear functional response) have to be guessed from the equation but are not explicitly mentioned.

Finally, I have the feeling that there are too many important information that have to be checked in appendix. In my opinion the model, which is the core of the article, should be described in the Methods not in supplementary. Showing the equations without parameters description is almost useless as readers do not have any idea of what they represent. The paper should be roughly understandable without the appendix I would say, here without the model description, it is not the case.

Equation 2 & 3: I spent a long time trying to understand why authors divided R_i by p_i in the functional response. This was not clear for me, even after reading Valdovinos *et al.* (2013). I guess it is because p_i is already included in V_{ij} . This kind of stuffs could be explained explicitly to the readers, to avoid them to get headache :)

Equation 3: Why do not use a classical functional response of type 2 for saturating the production of resources? As model behaviour is already described with this formulation, I understand that authors stucked t this choice, but I wonder if there is specific argument behind this choice.

Line 385: Why 121? where is that from? what are the parameter combination that is tested?

Lines 395-406: Authors performed motifs counts that are not corrected by richness, while they show that species richness is different between cases with or without adaptive foraging. I guess the number of different motifs in each categories (Spc-Gen, Gen-Spc, etc.) increases with network size, isn't it?

Richer is the network, more we expect motifs 1 and 3, because they are based on logical condition "at least", while we expect less motifs 1 and 4, because they are base on logical condition "only". This is exactly the pattern we observe between simulations with adaptive foraging (more motifs 1 and 3) and simulations without adaptive foraging (less motifs 1 and 3, more motifs 2 and 4), that could be explained completely by difference in species richness.

Thus, I wonder what do authors think about the possibility that difference sin motifs count is driven by difference in species richness mainly?

Lines 414-424: I realized at the end that what authors called network is not obvious. Do they mean the interaction matrix defined by V_{ij} (then including species abundance), or something independent from species abundance? Could be good to clarify it from the beginning.

Discussion: limits of the model are never highlighted or discussed, while I think it is always a good exercise and having the potential limits clearly written in the discussion helps the reader to link this work with previous one.

Table S1: last column of Table S1 is called "Motif frequency after establishments" where I guess after establishments is synonym of "after the subsequent colonization event" (line 127). I would suggest being very constant in the terminology used to help the readers, especially to maintain the word "subsequent" everywhere it is needed, to distinguish first colonization events than subsequent ones.

Appendix S2: if mortality is per capita, then it should be $\text{time}^{-1} \text{individuals}^{-1}$, isn't it?

Moreover, if parameters are drawn in gaussian distribution (that is not indicated), that means that negative values are possible, while it does not seem rational. How do authors deal with that?

References

- Bogdziewicz, M., Steele, M.A., Marino, S. & Crone, E.E. (2018). Correlated seed failure as an environmental veto to synchronize reproduction of masting plants. *New Phytol.*, 219, 98–108.
- Duchenne, F., Fontaine, C., Teulière, E. & Thébault, E. (2021). Phenological traits foster persistence of mutualistic networks by promoting facilitation. *Ecol. Lett.*, 24, 2088–2099.
- Guimarães, P.R., Pires, M.M., Jordano, P., Bascompte, J. & Thompson, J.N. (2017). Indirect effects drive coevolution in mutualistic networks. *Nature*, 550, 511–514.
- Higashi, M. & Nakajima, H. (1995). Indirect effects in ecological interaction networks I. The chain rule approach. *Math. Biosci.*, 130, 99–128.
- Nakajima, H. & Higashi, M. (1995). Indirect effects in ecological interaction networks II. The conjugate variable approach. *Math. Biosci.*, 130, 129–150.
- Pires, M.M., O'Donnell, J.L., Burkle, L.A., Díaz-Castelazo, C., Hembry, D.H., Yeakel, J.D., *et al.* (2020). The indirect paths to cascading effects of extinctions in mutualistic networks. *Ecology*, 101, e03080.
- Valdovinos, F.S., Espanés, P.M. de, Flores, J.D. & Ramos-Jiliberto, R. (2013). Adaptive foraging allows the maintenance of biodiversity of pollination networks. *Oikos*, 122, 907–917.

REVIEWER COMMENTS

Reviewer #1 (Remarks to the Author):

Review of "The role of intra-guild indirect interactions in assembling plant-pollinator networks," by Dritz et al.

The authors investigate the role of intra-guild indirect interactions on plant-pollinator network assembly using motif analysis and a colonization model that incorporates adaptive foraging. They find that adaptive foraging promotes the coexistence of specialists and generalists within the same guild and that colonizers tend to form intra-guild indirect interactions with species of opposite niche breadth (specialist colonizers with generalist incumbents for plants and generalist colonizers with specialist incumbents for pollinators).

The manuscript is generally well written, especially the discussion, and the core ideas---motif analysis for studying indirect interactions and the effects of adaptive foraging on community assembly---are novel and interesting. My main critique is that the study is heavily theoretical/modelling/simulation-based and would really benefit from additional analyses involving empirical networks. I also have some other suggestions for improving the manuscript.

Response: We thank the reviewer for the constructive criticism and useful suggestions that have greatly improved the quality of our manuscript. We have incorporated them all in the new version of our manuscript except for the additional analysis involving empirical plant-pollinator networks. Our next response explains why our motif analysis cannot be applied to static networks, and why current temporal plant-pollinator networks data are not well-suited for our analysis. We connect our results with empirical findings in the discussion section, which is the best we can do in terms of data-theory integration given the limitations of current empirical data. After all, one main reason we use theoretical approaches to investigate important questions in ecology is that empirical research has temporal, spatial, and other scale-related limitations. We elaborate on this in the discussion section (L450-461).

-- Tighter integration of empirical networks in analyses

The authors provide some comparison of their theoretical work to empirical networks (L255, L260, etc.), but these analyses are relatively limited and appear as more of an afterthought. While I like the overall modelling approach and find the theoretical results interesting, it will be important to see whether the findings translate to empirical networks. I appreciate that it is not possible to study the assembly of these empirical networks, but can the authors do something like

provide a breakdown of motif frequencies in empirical networks? Right now, the comparison centers on three broad network properties (richness, connectance, and nestedness, see Fig. 6) and the results themselves are not especially clear and convincing and, moreover, do not provide any additional support to the main findings regarding intra-guild competition and higher-level interaction patterns.

Response: We agree with the reviewer that the comparison with empirical networks based on broad properties (richness, connectance, and nestedness) does not provide strong support for our main findings of how intra-guild indirect interactions of colonizers contribute to higher-level interaction patterns in assembling pollination networks. For this reason, that figure was moved to the Supplementary Information (see new Figure S1).

However, to the best of our knowledge, there is no empirical data on assembly of plant-pollinator networks both at the scale of 150 years and at a fine enough temporal resolution (i.e., sampled at least every 3 years) to make a fair comparison with our motif results. Moreover, the motifs developed in this study are not meant to be a metric to characterize the structure of a static pollination networks. Rather, they are meant as a tool to investigate the transient dynamics of colonizer establishment via intra-guild indirect effects (L154-157). Specifically, our motifs focus on a focal species that is followed from its colonization to subsequent extinctions and colonizations. Therefore, providing a breakdown of motif frequencies in empirical networks (shown below for all 121 empirical networks used for new Figure S1) would be an erroneous way to use our motif analysis as there is no assembly information that can be tracked for each focal species. Therefore, we cannot include this analysis in the new version of our manuscript. We explain this limitation of current empirical data, which is also the strength of our study – as we can unveil an understanding of network assembly that is not possible to obtain via empirical research, at least with current data – in L450-461.

-- Additional temporal analyses

The authors record motif groups in sets of three subsequent time steps (L124) but then group the motif data for all colonizers, regardless of when they established in a network (L127). It would be very interesting to see an explicit breakdown of the distribution of these three-motif sequences (i.e., similar to information presented in Table 1). It would also be cool to see if/how motif distributions change during the colonization trajectory, e.g., early vs. middle vs. late stages.

Response: Following the reviewer’s suggestion, we extended our theoretical analysis to investigate whether motif distributions among attempted and established colonizers varied among early, mid, and late state colonizers by adding Fig S2, which more intuitively illustrates the information previously presented in old Table 1 (now new Table S2). We found that the distribution of motifs among attempted colonizers does not vary significantly between these groups. However, the establishment rate across all motif groups decreased over time, meaning that early colonizers established at a higher rate than later colonizers which is consistent with the empirical evidence we referred to in the discussion. We also expanded new Table S3 (prior Table S1) to more explicitly show the cumulative distribution of motifs (across all colonizers in each assembly model) at each of the three subsequent timesteps, that is: Motif frequency after

establishment, motif frequency after extinctions, and motif frequency after subsequent establishment.

-- Remind the reader of the four questions

I like the four research questions (L76) but found it difficult to remember each question when they were referred to simply by Q1 etc. This was especially the case when reading the section "Core motifs produced by plant species" (L142). It would be helpful if sentences could be rewritten to remind readers of each question as the answers are presented.

Response: Agreed. Sentences were rewritten throughout the results to clarify which question each result addresses. For example, for the section "Core motifs produced by plant species with adaptive foraging" these questions can be found in L193, L196, L200, and L204.

-- Clarify what information is presented in Table 1 and Fig. 4

I did not find the information in Table 1 very easy to interpret. Is the establishment rate over all 121 simulations? Is such information worth presenting by model (W/ AF etc.) rather than broken down by plants vs. pollinators? Which rates are meant to sum to 1? The bottom line is that I did not find Table 1 very effective. Also, it wasn't clear to me if Fig. 4 (which I did find intuitive) presented the same/similar information to Table 1.

Response: We agree with the reviewer that Table 1 in the previous version of our manuscript was not easy to interpret so we moved it to the supplementary information (Table S2 in the revised manuscript) and replaced it with a new figure (Fig. 4 in the revised manuscript) which more intuitively shows the same information. To clarify, Fig. 4 in the previous manuscript showed the distribution of motifs following extinctions and subsequent establishment while Table 1 in the previous manuscript showed the distribution of motifs prior to extinctions and subsequent establishment. Captions of new Table S2, new Figure 4, and new Figure 7 clarify this difference.

-- Consider specialists as having more than one interaction

I would be curious to know if it would be possible to perform the analysis with specialists defined by a diet breadth greater than one (L218)? Could it also work if specialists had a potential diet breadth greater than one but only realized one interaction at a time? In a similar vein, do the authors think results would change significantly if species were categorized into specialists, generalists, and super-generalists (with motifs reflecting this categorization)? Such an extension would help bring the work closer to empirical systems, in which diet breaths follow

distributions that are much more complicated than just specialist having 1 interaction and generalists having > 1 interactions. Also, in simulations the number of interactions of a generalist is drawn from a uniform distribution (L388)---would results change if other distributions were used, particularly ones that more closely match empirical degree distributions?

Response: We thank the reviewer for making these comments, which helped us see that we needed to clarify that our assembly model actually reproduces all the attributes of empirical networks the reviewer points out. We address this set of comments by adding more panels to new Figure S1. We explain how the new panels of Fig. S1 answer each comment in the last paragraph of this response.

The definition of specialist plant and pollinator species having only one interaction while generalist species having more than one, directly connects with qualitatively different behaviors of the Valdovinos et al's (2013) model. On the one hand, specialist plant species offer the most exclusive floral rewards to the pollinator species that visits them and specialist pollinator species offer the highest quality of visits to the one plant species they visit but cannot adaptively re-arrange their foraging efforts because they only interact with one plant species. On the other hand, the rewards offered by the generalist plant species are shared by several pollinator species (causing exploitative competition) and generalist pollinator species visit several plant species, diluting their conspecific pollen. Given these very distinct differences in model behavior between species having one interaction versus having more than one interaction, our motif analysis based in such a distinction is very effective in capturing the effect of intra-guild indirect interactions on community assembly.

There is only a quantitative difference in model behavior between generalist and super-generalist pollinator species (e.g., between 3 and 20 interactions), that is, super-generalist pollinators have the greatest niche flexibility and as a result become the most quantitatively specialized. However, distinguishing between generalists and super-generalists does not increase our ability to detect intra-guild indirect interactions.

In summary, the distinction between specialists and generalists in our analysis is the most effective way to detect intra-guild indirect interactions in a very complex set up (assembly + population dynamics + adaptive foraging), which is a strength of our work. That is, it is a smart distinction to conduct an analysis that disentangles simple dynamic patterns from a very complex dynamical process. Our assembly model still produces degree distributions (including the full array of specialist to super-generalist species) observed in empirical networks that the reviewer is referring to (see new Fig S1A,B). Moreover, the species that colonize with only one interaction usually end up with more than one interaction throughout the assembly process. Thus, in essence we

are already modeling species that have “a potential diet breadth greater than one but only realized one interaction at a time” rather than “true specialists” as the reviewer suggests. Finally, to the best of our knowledge, there is no empirical data on the distribution colonizers’ degrees when they enter a network. Therefore, we believe that the uniform distribution is the most parsimonious.

-- Describe how results for plant-focused motifs fit with pollinator-focused motifs

Results for pollinators are by-and-large considered *separately* from results for plants. I would like to see a discussion on how the results regarding pollinators (e.g., generalist colonizers with specialist incumbents) and plants (e.g., specialist colonizers with generalist incumbents) might emerge *together* from a modelling/technical standpoint; this could go around L253.

Response: We followed the reviewer’s suggestion by discussing in L406-414 how the core “Spec-Gen” and “Gen-Spec” motifs for plants and pollinators, respectively, can emerge together. That is, the specialist plant colonizers in motif group “Spec-Gen” can provide exclusive resources to generalist pollinator colonizers in motif group “Gen-Spec” who can in turn perform high quality visits by quantitatively specializing on those plants. This advantageous “mega-motif” combining plants’ “Spec-Gen” and pollinators’ “Gen-Spec” motifs is the most likely to emerge from assembly given its positive, direct effects on species of the opposite guild. Moreover, this “mega-motif” would lead to even more nested networks.

-- Fig. 1. I don't know what a "passing" interaction is? Is there a better term that can be used instead? Maybe "transient" or "temporary"? Or perhaps add a definition in the caption.

Response: Changed “passing” to “extinct”. The dashed line is meant to indicate which interactions disappear as a result of a species becoming extinct.

-- L109. Add a short explanation of the "dynamic species pool" and how it works.

Response: This phrase was deleted. The intention was to say that colonizing species (identified by characteristics such as their foraging efficiency or nectar production rate) are randomly generated rather than originating from a finite regional species pool. However, on further investigation, the phrase “dynamic species pool” indicates in prior literature that eco-evolutionary feedbacks are influencing species’ characteristics which is not the case in this model. Thus, we deleted the phrase to avoid any confusion with prior uses of this phrase.

-- L122. It would be helpful to have an example of how "colonizers' interactions transform from

one motif group to another." I had a hard time imaging what this would look like.

Response: An example was added to L165-167. That is “For instance, if a specialist colonizer belonging to motif group “Spec-Spec” transformed to motif group “Spec-Gen”, this would indicate that all intra-guild indirect specialists went extinct”.

-- Fig. 2. I think the figure title would be clearer if it was, "...each motif group." Also, personally I would find "Spec" easier to read than "Spc".

Response: Both suggestions were incorporated in what is now Figure 3 (old Fig 2).

-- Fig. 3. Because most plant-pollinator networks are drawn with plants as the lower guild and pollinators as the upper guild, it would be easier for me to follow if figures reflected this convention.

Response: Thank you for the suggestion, we flipped the motifs in the figure to conform with the convention of plants as the lower guild.

-- Fig. 6. I did not understand what was done based on the sentence (L272) beginning, "We performed a one-sided..." Consider expanding the technical description.

Response: We added an Appendix S3 to explain which structural metrics we used in our analysis and which statistical tests were performed. We performed Welch tests to statistically evaluate the hypothesis that networks assembled from the model with adaptive foraging are richer, less connected, have a higher pollinator:plant ratio, and are more nested than networks assembled from the model without adaptive foraging. The Welch tests were paired to compare networks populated by specialist plants and specialist pollinators at the same probability. Additionally, we performed Welch test to statistically evaluate the hypothesis that the connectance, plant:pollinator ratio, and richness of empirical networks is significantly different from simulated networks produced from our assembly model with adaptive foraging.

-- L327. I found the argument involving *A. mellifera* and *Bombus spp.* confusing since the example with *Bombus spp.* doesn't seem to support the general thesis of the paragraph.

Response: Agreed, thank you for pointing this out. We edited the text to acknowledge that *A. mellifera* follows the trend in our model but *Bombus spp.* do not (L386-389).

Reviewer #2 (Remarks to the Author):

General comment

Dritz et al. developed a dynamic model to describe the assembly of mutualistic network, a surprisingly overlooked aspect of mutualistic networks. By including adaptive foraging in this article, they provided an innovative theoretical approach of the question, allowing them to account for the plasticity of interactions, that is often neglected. By combining this model with a motifs approach, they could track the impact of colonization on the network over the assembly process. This article is definitely a nice piece of science that provides a new result about the dynamics of invasion in a network context. The introduction and discussion are well written and clear. I have no major comments about that manuscript. I have two main points of discussion I would like to raise: 1) about the use of motifs and possible bias and limits; 2) for the moment big parts of the methods are quite obscure and require a lot of energy from the reader to be understood because we need to read previous articles about the model, and the format (methods at the end) does not help. I detail a bit these points in the following paragraphs. Otherwise figures and results could be a bit clearer, but as I said above, these are no major issues and this article is really a worthy reading, thanks to the authors for that.

Response: We thank the reviewer for the constructive criticism and useful suggestions that have greatly improved the quality of our work. We have addressed them all in the new version of our manuscript. We paid special attention to making the Methods much easier to follow, and provided all the information needed to understand them.

Specifically, we brought all the methods from previous papers and our supplementary material to the new version of our Methods section.

Authors focused on indirect effects that propagate through paths of length 2 only, so the shortest indirect effects possible, but are never highlighted, while it would be nice to discuss this point. However, many studies have shown that in diverse networks like those studied by authors, an important part of indirect effects propagate through long-paths (Higashi & Nakajima 1995; Nakajima & Higashi 1995; Guimarães et al. 2017; Pires et al. 2020). The same studies using the Jacobian matrix of dynamical systems to study indirect effects over all these possible paths (Nakajima & Higashi 1995), that in contrast to motif approach allow to integrate these effects and measure their strength, including abundances and interaction strengths to calculate propagation from species to species. Thus, I wonder why did authors prefer motifs approach relative to this method? About motifs analyses, also see my comment for lines 395-406, that is, I think, of importance.

Response: Thank you for the suggestion as well as the great references. We added a discussion comparing the relative strengths of each method in L42-49. Previous studies have used mathematical methods to determine the strength of indirect effects through short and long paths by accounting for positive and negative feedbacks. Network motifs function as a complimentary tool to those methods by establishing a connection between indirect effects and network structure. The trade-off is that considering indirect effects through longer paths requires larger and more complex motifs which are more difficult to interpret. We chose to only consider indirect effects among species sharing a direct mutualistic partner (a path of length 2) to detect clear dynamic patters in a relatively complex set up (assembly + population dynamics + adaptive foraging). However, defining more complex motifs to capture indirect effects through longer paths is an avenue for future research.

The model is not understandable without referring to other papers, because parameters and variables are not described at all in the main Methods and partially in Appendix S2, which is, I think, a problem to understand the study. Some important equations are missing, for example the equation linking V_{ij} to α_{ij} and to species abundances. To summarise, in absence of further explanations and description the equations presented in Methods are useless. Nerveless, once the reader has done the effort to read previous paper (Valdovinos et al. 2013), this model is definitely a nice piece of work and this paper a nice piece of science. But more efforts are needed in explaining (again) the model in the present study.

Response: Thank you for pointing this out, we brought all the methods from previous papers and our supplementary material to the new version of our Methods section, which greatly improves the readability of our manuscript. We added descriptions of each parameter to the text of the methods. Additionally, we added equations for V_{ij} (Eq. 5), σ_{ij} (Eq. 6), and γ_i (Eq. 7). Lastly, we expressed each equation in relation to the functional response to ease model interpretation. The functional response is defined in Eq. 8.

Point-by-point comments

Line 38: I do not know this paper but by quickly reading through I do not see how ref. 20 (Bogdziewicz et al. 2018) is linked to this point. I might have missed something, but good to check if it is the right reference.

Response: Agreed, this reference is related to the reproductive benefits of indirect effects but not assembly so this citation was moved to the last line in the paragraph.

Lines 75-79: here and in the following parts, I started to have a problem of terminology to fully

understand. In question 1 a colonizer is a species that attempt to colonize the system, regardless the success of the colonization. But in following questions, a colonizer becomes a successful colonizer. I think it is important to distinguish both. When authors study motifs, do they present the result sonly for successful colonizer or all colonizers?

Edit: answer to that question is present lines 407-413, but since Methods are at the end might be good to find a way to clarify it in the main text.

Response: In our motif analysis we only consider colonizers that successfully establish in the network, we added this to L85-86. In addition, we added the terminology “attempted colonizer” and “established colonizer” to more clearly distinguish between when we’re referring to which group.

Lines 78-79: at this point, it is not intuitive what is behind question 3 for me. I struggle to see where this question goes, which kind of mechanism it aims to explore (competitive exclusion? Niche partitioning?) and what are the assumptions of authors.

Response: Questions 2 and 3 were rephrased to: “Do colonizers competitively exclude intra-guild indirect specialists” and “Do colonizers facilitate the establishment of subsequent colonizing intra-guild indirect specialists?”, respectively, to show that the questions complement one another. The goal of these questions is to determine whether indirect effects change in strength and nature (facilitative or competitive) as colonizers become established. Through these questions we can determine the ultimate effect of colonizers on intra-guild indirect specialists.

Lines 111-113: without further explanation, this sentence is very enigmatic. It could be either developed here, or removed from here and developed in Methods.

Response: Agreed, this sentence was moved to methods and elaborated on in L506-514.

Line 116: before entering in motif description, I really missed basic information on network size at the end of the process. So, it starts by 3X3 networks, and could end with 150X150 networks if all colonizing species survived and there is no extinction. Knowing the average network size at the end of simulation would be helpful to understand motifs analyses.

Edit: After continuing reading I found richness and connectance plots latter, but I really think it misses some description of the dynamics and end point of the simulation before describing the motifs part, otherwise it is really abstract. Authors could have a first plot at this stage that describe the dynamics of simulations and number of successful colonization and extinctions. Something in the spirit of what is below.

Response: Thank you for this excellent suggestion. We added new Fig. 2 to describe the trajectory of network structure (richness, connectance, and nestedness) in each assembly model as well as the number of species establishments and extinctions per simulation to give readers context for the following motif analysis.

Lines 124-127: I should admit that for me it was not clear all long the article I had the feeling I had to guess when authors used the motifs count calculated at the moment of the colonization, after the extinctions, or after the subsequent colonization event. I think it could be labelled more clearly.

Response: Agreed. This information was clarified in the legends of Fig. 4, Fig. 7, and Table S2 in the revised manuscript. In addition, we clarified this throughout the written results.

Fig. 2: I think that a bit of colours could help to understand better the figure.

For example:

Fig. 2 | Schematic representations of each motif. Each motif characterizes the niche breadth (specialist or generalist) of the focal colonizing species (specie highlighted by the box) and the niche breadth of species that share its mutualistic partner(s), as follows: 1) “Spc - Spc”, specialist colonizer (blue) that interacts indirectly with at least one specialist in its guild (orange): ...

Response: Thanks for this very useful suggestion. We added colors to this figure as suggested by the reviewer in what is now new Figure 3.

Fig. 3: in C, it took me time to guess that authors mean “No subsequent specialist establishment”, as subsequent is missing.

Response: Agreed. We changed “No specialist establishment” to “No subsequent specialist establishment” in Fig. 5, Fig. 6, Table S3, and kept such wording consistent throughout the entire manuscript.

It is just a convention that can be changed but having the plants as the top guild of the bipartite network is not intuitive (they are often represented as the bottom guild, with pollinators as top guild). Since the paper is complex, I would stick to classic norms as much as possible to keep the readers focusing on important complexity.

Response: Agreed. We flipped the motifs to conform with the convention of having plants as the lower guild.

Table 1: so this table correspond to motifs account at the moment of colonizer’s arrival? Could be good to be very clear about that.

Response: The reviewer is correct. We added the suggested information to the legend of the new Fig 4 and Table S2 in the revised manuscript.

Lines 285-288: Isn’t table 1 instead of Figure 4 that shows this result?

Response: Yes, thank you for pointing this out. We changed the reference from Fig. 4 in the previous manuscript to Fig. 4 and Table S2 in the revised manuscript.

Lines 293-297: I find the wording “mutualisms mediate species coexistence” a bit weird. I would have said “adaptive foraging mediates species coexistence”, as without adaptive foraging mutualism does not maintain high coexistence.

Response: We changed the phrasing to: “mutualisms can mediate species coexistence through niche differences enabled by adaptive foraging” (L353-356) because the focus of this phrase is on the role of mutualisms, but it is correct that we’ve shown that mutualisms can destabilize species coexistence in the absence of adaptive foraging.

Lines 346-350: About the emergence of nestedness through indirectly connected species it could be worthy to link this result to similar results found when introduced a phenological structure in

mutualistic networks (Duchenne et al. 2021). Like adaptive foraging, phenological structure also protects specialist from extinctions through indirect effects, and thus promote nestedness and stable coexistence of a higher diversity (Duchenne et al. 2021).

Response: Thank you for sharing this reference. We added the results of this paper to our discussion of the emergence of nestedness (L423-427). Duchenne et al 2021 expanded on Bastolla's model to investigate how network structure arising from morphology and phenology influences the nature of indirect interactions. They found that morphology increases indirect competition while phenology increases indirect facilitation which produces more diverse and nested networks in which intra-guild indirect generalists and specialists can coexist. While our assembly model is dominated by indirect competition, we observe the emergence of nested networks because adaptive foraging alleviates the strength of indirect competition to maintain species coexistence. Importantly, however, adaptive foraging does not reverse indirect effects to make them facilitative.

Lines 376 – 380: I could not find the meaning and units of γ_i , σ_{ij} , V_{ij} .

Variable names are not described. Often for readability variables are represented with capital letters, while parameters with lowercase letters, I think following this convention would help to understand quickly the equations, that are here really hard to understand, especially without description.

Basic assumptions of the model (e.g. linear functional response) have to be guessed from the equation but are not explicitly mentioned.

Finally, I have the feeling that there are too many important information that have to be checked in appendix. In my opinion the model, which is the core of the article, should be described in the Methods not in supplementary. Showing the equations without parameters description is almost useless as readers do not have any idea of what they represent. The paper should be roughly understandable without the appendix I would say, here without the model description, it is not the case.

Response: Agreed, we added a description of all parameter values to the text of the methods and added the missing equations for visitation, visit quality, and seed recruitment (Eq. 5-8).

Equation 2 & 3: I spent a long time trying to understand why authors divided R_i by p_i in the functional response. This was not clear for me, even after reading Valdovinos et al. (2013). I

guess it is because p_i is already included in V_{ij} . This kind of stuffs could be explained explicitly to the readers, to avoid them to get headache :)

Response: Thank you for the suggestion, we rewrote the model's equations in terms of the functional response which is defined in Eq. 8.

Equation 3: Why do not use a classical functional response of type 2 for saturating the production of resources? As model behaviour is already described with this formulation, I understand that authors stuck to this choice, but I wonder if there is specific argument behind this choice.

Response: We chose to use a type 1 functional response because the production of floral resources saturates in Eq. 3 through the phi parameter (L496-498).

Line 385: Why 121? where is that from? what are the parameter combination that is tested?

Response: Clarified why 121 simulations were run for each model in L108-111.

Lines 395-406: Authors performed motifs counts that are not corrected by richness, while they show that species richness is different between cases with or without adaptive foraging. I guess the number of different motifs in each categories (Spc-Gen, Gen-Spc, etc.) increases with network size, isn't it? Richer is the network, more we expect motifs 1 and 3, because they are based on logical condition "at least", while we expect less motifs 1 and 4, because they are based on logical condition "only". This is exactly the pattern we observe between simulations with adaptive foraging (more motifs 1 and 3) and simulations without adaptive foraging (less motifs 1 and 3, more motifs 2 and 4), that could be explained completely by difference in species richness.

Thus, I wonder what do authors think about the possibility that difference in motifs count is driven by difference in species richness mainly?

Response: This is a really interesting point. The distribution of motifs among attempted colonizers is definitely sensitive to network structure (richness, connectance, and nestedness) which is described in L437-447. To evaluate how strongly motifs are biased by richness in our study, we compared the distribution of motifs among attempted colonizers between early, mid, and late-stage colonizers (which are introduced to networks of varying size) in Fig. S2. We found that the distribution of motifs does not vary significantly between these groups indicating that richness is not strongly biasing our analysis. Rather, variation in the distribution of motifs among attempted colonizers between assembly models is due to the number of specialists

populating the network (influencing network connectance) and whether those specialists are connected to generalist hubs (influencing network nestedness). This speaks to the feedback of network assembly. While the process of colonizer motif establishment produces network structures, network structures influence the process of colonizer motif establishment.

Lines 414-424: I realized at the end that what authors called network is not obvious. Do they mean the interaction matrix defined by V_{ij} (then including species abundance), or something independent from species abundance? Could be good to clarify it from the beginning.

Response: We are referring to binary network structure (independent from species abundance) which was clarified in Appendix S3 and at the beginning of the results where we give an overview of the simulated networks (L133).

Discussion: limits of the model are never highlighted or discussed, while I think it is always a good exercise and having the potential limits clearly written in the discussion helps the reader to link this work with previous one.

Response: Agreed. We have added a discussion of the limitations of our model in L433-447. We highlighted two main limitations. First, our motifs can only capture indirect effects through a path length of 2. Second, the frequency of motifs among attempted colonizers varies based on the structure of the network they are colonizing. For instance, motif groups “Spec-Spec” and “Gen-Spec” are defined by having *at least one* intra-guild indirect interaction with a specialist. Therefore, they will be more common among attempted colonizers in: 1) richer networks where species have more indirect partners, and 2) less connected and more nested networks where there are more specialists which are connected to generalist hubs. “Spec-Spec” and “Gen-Spec” Those network structures because

Table S1: last column of Table S1 is called “Motif frequency after establishments” where I guess after establishments is synonym of “after the subsequent colonization event” (line 127). I would suggest being very constant in the terminology used to help the readers, especially to maintain the word “subsequent” everywhere it is needed, to distinguish first colonization events than subsequent ones.

Response: Agreed. We changed the column title in our new Table S3 (old Table S1) to “Motif frequency after subsequent establishment” and used the terminology “subsequent colonization event” throughout the text.

Appendix S2: if mortality is per capita, then it should be time⁻¹ individuals⁻¹, isn't it? Moreover, if parameters are drawn in gaussian distribution (that is not indicated), that means that negative values are possible, while it does not seem rational. How do authors deal with that?

Response: Parameters were drawn from uniform distributions which were included in the description of Table S1. We have assertions in the code to check for negative values which will cause the simulation to stop. We checked for consistency in all units presented in Table S1 (equations must have final units of individuals area⁻¹ time⁻¹) and the mortality parameter is time⁻¹. Those units are consistent with all previous publications of the model.

References

- Bogdziewicz, M., Steele, M.A., Marino, S. & Crone, E.E. (2018). Correlated seed failure as an environmental veto to synchronize reproduction of masting plants. *New Phytol.*, 219, 98–108.
- Duchenne, F., Fontaine, C., Teulière, E. & Thébault, E. (2021). Phenological traits foster persistence of mutualistic networks by promoting facilitation. *Ecol. Lett.*, 24, 2088–2099.
- Guimarães, P.R., Pires, M.M., Jordano, P., Bascompte, J. & Thompson, J.N. (2017). Indirect effects drive coevolution in mutualistic networks. *Nature*, 550, 511–514.
- Higashi, M. & Nakajima, H. (1995). Indirect effects in ecological interaction networks I. The chain rule approach. *Math. Biosci.*, 130, 99–128.
- Nakajima, H. & Higashi, M. (1995). Indirect effects in ecological interaction networks II. The conjugate variable approach. *Math. Biosci.*, 130, 129–150.
- Pires, M.M., O'Donnell, J.L., Burkle, L.A., Díaz-Castelazo, C., Hembry, D.H., Yeakel, J.D., et al. (2020). The indirect paths to cascading effects of extinctions in mutualistic networks. *Ecology*, 101, e03080.
- Valdovinos, F.S., Espanés, P.M. de, Flores, J.D. & Ramos-Jiliberto, R. (2013). Adaptive foraging allows the maintenance of biodiversity of pollination networks. *Oikos*, 122, 907–917.
-

Reviewers' Comments:

Reviewer #1:

Remarks to the Author:

Review of revised "The role of intra-guild indirect interactions in assembling plant-pollinator networks," by Dritz et al.

I am Reviewer 1 from the first round of reviews. I have read the revised manuscript and the authors' response to reviews. I appreciate the thoughtful replies to my comments and understand the rationale when changes were made and when when they were not. I believe the work to be technically sound and I like the theoretical ideas and methods presented in the manuscript. However, I still would have liked to have seen greater integration with empirical data, although I understand how this is challenging. Perhaps the authors could provide more details on how their ideas could be tested with empirical data that could be collected on time scales of years rather than decades, beyond just the text added in the closing paragraph. In conclusion, while I cannot offer wholehearted support for publication, I would not stand in the way.

REVIEWERS' COMMENTS

Reviewer #1 (Remarks to the Author):

Review of revised "The role of intra-guild indirect interactions in assembling plant-pollinator networks," by Dritz et al.

I am Reviewer 1 from the first round of reviews. I have read the revised manuscript and the authors' response to reviews. I appreciate the thoughtful replies to my comments and understand the rationale when changes were made and when when they were not. I believe the work to be technically sound and I like the theoretical ideas and methods presented in the manuscript. However, I still would have liked to have seen greater integration with empirical data, although I understand how this is challenging. Perhaps the authors could provide more details on how their ideas could be tested with empirical data that could be collected on time scales of years rather than decades, beyond just the text added in the closing paragraph. In conclusion, while I cannot offer wholehearted support for publication, I would not stand in the way.

Response: Thank you for the suggestion to provide more detail on how the key results of the paper could be tested empirically. We added this to the discussion L354-375.